# The dynamic recruitment of TRBP to neuronal membranes mediates dendritogenesis during development

Anna Antoniou[1,*] , Sharof Khudayberdiev[1], Agata Idziak[1], Silvia Bicker[1] , Ralf Jacob[2] & Gerhard Schratt[1,†,**]

## Abstract

MicroRNAs are important regulators of local protein synthesis during neuronal development. We investigated the dynamic regulation of microRNA production and found that the majority of the microRNA-generating complex, consisting of Dicer, TRBP, and PACT, specifically associates with intracellular membranes in developing neurons. Stimulation with brain-derived neurotrophic factor (BDNF), which promotes dendritogenesis, caused the redistribution of TRBP from the endoplasmic reticulum into the cytoplasm, and its dissociation from Dicer, in a $Ca^{2+}$-dependent manner. As a result, the processing of a subset of neuronal precursor microRNAs, among them the dendritically localized pre-miR16, was impaired. Decreased production of miR-16-5p, which targeted the BDNF mRNA itself, was rescued by expression of a membrane-targeted TRBP. Moreover, miR-16-5p or membrane-targeted TRBP expression blocked BDNF-induced dendritogenesis, demonstrating the importance of neuronal TRBP dynamics for activity-dependent neuronal development. We propose that neurons employ specialized mechanisms to modulate local gene expression in dendrites, via the dynamic regulation of microRNA biogenesis factors at intracellular membranes of the endoplasmic reticulum, which in turn is crucial for neuronal dendrite complexity and therefore neuronal circuit formation and function.

**Keywords** BDNF-induced dendritogenesis; endoplasmic reticulum; isomiR; miR-16; pre-miRNA processing
**Subject Categories** Membrane & Intracellular Transport; Neuroscience; RNA Biology

See also: **E Hornstein** (March 2018)

## Introduction

The local regulation of the dendritic proteome is crucial for the morphological and functional heterogeneity of neuronal dendrites and therefore neuronal circuit formation and plasticity [1–3]. MicroRNAs (miRNAs) are small, non-coding RNAs that regulate protein synthesis by binding to the 3′ untranslated regions (UTRs) of mRNA targets and inhibiting their translation [4,5]. Due to their rapid and reversible effects on protein translation [6–8], miRNAs are ideal candidates for the activity-dependent regulation of the local dendritic proteome. Indeed, many miRNAs were shown to modulate neuronal morphogenesis, for example, by regulating components of actin-regulatory pathways [9–12], transcription factors [13,14], and other plasticity-related proteins [15,16]. Whether and how miRNA production is confined at the subneuronal level is so far unknown.

MiRNAs are derived from stem-loop, precursor miRNA transcripts (pre-miRNAs) that are exported from the nucleus [17–19] and in some cases transported into neuronal dendrites [20–22]. The endoribonuclease Dicer cleaves pre-miRNAs into approximately 22-nucleotide-long microRNA duplexes [23]. One of the strands of the miRNA duplex is then loaded onto Argonaute family of proteins (Ago), which are the major catalysts of the miRNA-induced silencing complex (RISC) [24,25]. The RISC-loading complex (RLC) consists of Dicer and its co-factors TRBP (*HIV1-TAR RNA Binding Protein*) and/or PACT (*Protein ACTivator of the interferon-induced protein kinase*) and Ago [26–29]. TRBP and PACT are protein orthologues that bind to double-stranded RNA and play a role in RNA interference and cellular stress responses by regulating protein kinase R (PKR) [30]. Although TRBP and PACT are known components of the RLC, their precise role in pre-miRNA processing is still unclear.

Several publications have demonstrated the association of miRNA-regulating proteins at intracellular membranes. For instance, it was shown that Ago associates at intracellular endosomes [31–33]. Furthermore, previous reports suggest that the RISC forms at the rough endoplasmic reticulum (rER) [34,35], which is a site for

1  Institute for Physiological Chemistry, Biochemical-Pharmacological Center Marburg, Philipps-University of Marburg, Marburg, Germany
2  Department of Cell Biology and Cell Pathology, Philipps-University of Marburg, Marburg, Germany
   *Corresponding author. Tel: +49 64212862289; E-mail: anna.antoniou@staff.uni-marburg.de
   **Corresponding author. Tel: +41 44 633 81 32; E-mail: gerhard.schratt@hest.ethz.ch
   †Present address: Department of Health Science and Technology, Laboratory of Systems Neuroscience, Swiss Federal Institute of Technology, Zurich, Switzerland

protein production and homeostasis. Indeed, many miRNAs were shown to associate with neuronal polyribosomes [36,37], suggesting that this may also be the case in neurons.

Here, we show that the RLC associates at the neuronal ER in the soma and dendrites of developing rat primary neurons. We observed that the subcellular localization of TRBP is dynamic following brain-derived neurotrophic factor (BDNF) stimulation, which affects the production of miR-16-5p. We further show that BDNF inhibits the Dicer–TRBP interaction and pre-miR16 processing by TRBP-containing complexes. Finally, miR-16-5p is a negative regulator of dendritic complexity and mediates BDNF-induced dendritogenesis by regulating the translation of the BDNF mRNA itself. Our results point towards a novel neurobiological mechanism, whereby the activity-dependent regulation of TRBP localization at ER membranes mediates BDNF-induced dendritogenesis by regulating the production of miR-16-5p.

## Results

### The RISC-loading complex associates at the neuronal endoplasmic reticulum

We first sought to characterize the subcellular distribution of RISC and RLC proteins in neurons. Postnuclear homogenates from developing rat primary cortical cultures were loaded underneath a continuous non-ionic and iso-osmotic density gradient consisting of iodixanol. We obtained efficient separation of the smooth and rough ER, in fractions 1–3 and 4–6, respectively, and found that a large majority of TRBP and PACT co-sediment with the rER markers, ribophorin I, Climp63, and ribosomal protein S6 (RPS6), that also consist of the mitochondrial marker Tim23 (fractions 4–6; Fig 1A). A portion of Dicer, Ago1, and Ago2 as well as markers of RNA granules such as FMRP (*Fragile × mental retardation protein*) and Staufen 2 also co-sedimented in rER fractions. In contrast, the RISC component GW182 was absent from the rER, which is in line with a previous publication in *Drosophila* [35], but co-sedimented in light fractions with the late endosomal marker Lamp2, consistent with a previous publication in non-neuronal cells [32]. High-density fractions 8–10 consisted of RPS6, FMRP, Staufen 2, and GW182, as well as RISC and RLC proteins, suggesting that these fractions correspond to RNA granules and RNA processing bodies (P-bodies). Notably, Dicer, Ago, and PACT, but not TRBP, were present in heavy fractions, which may indicate the presence of distinct RLC complexes. Moreover, the majority of Dicer is present in light fractions 1–3, corresponding to smooth ER as indicated by calnexin, which also consist of plasma membrane, endosomal markers as well as PACT (Fig 1A). Nevertheless, we could isolate Dicer along with TRBP, PACT, and Ago1 and Ago2 in ER microsomes purified using differential centrifugation (Fig EV1A, Appendix Fig S1A), suggesting that at least a proportion of Dicer localizes at the ER.

Using immunocytochemistry, we observed that endogenous TRBP, PACT, and Ago2 localize at perinuclear regions that partially overlap with mCherry-Climp63 in primary hippocampal neurons (Fig 1B). In addition, we detected partial co-localization of these proteins with mCherry-Climp63 at primary dendrites and dendritic branch points (inserts in Fig 1B). Further, GFP-TRBP and GFP-PACT co-localized with calnexin at perinuclear regions, providing

independent support for the ER localization of these proteins in neurons (Fig EV1B). We did not observe co-localization between endogenous TRBP and PACT with Mitotracker Deep Red, thus excluding the possibility that TRBP and PACT are present at mitochondria (Fig EV1C). Furthermore, HA-tagged GW182 did not co-localize with mCherry-Climp63 (Appendix Fig S1B), thereby confirming results obtained by density gradient fractionation (Fig 1A). As we could not find a reliable Dicer antibody for immunocytochemistry, we utilized total internal reflection fluorescence (TIRF) microscopy and imaged hippocampal neurons expressing GFP-Dicer and mCherry-Climp63. We set the penetration depth to 150 nm in order to visualize the ER at high signal-to-noise ratio. GFP-Dicer partially overlaps with mCherry-Climp63 at the cell soma and primary dendrites of hippocampal neurons (Fig 1C), further supporting our fractionation results.

We further examined the association of RLC proteins with intracellular membranes using sequential detergent extraction in young cortical neurons as well as primary astrocytes and Hek293T cells. The mild detergent digitonin was used to permeabilize the plasma membrane and extract the cytosolic fraction, followed by application of stronger detergents to extract all other intracellular membranes apart from the nucleus. In neurons, the majority of Dicer, TRBP, and PACT were present in membrane fractions, while Ago1 and Ago2 are equally distributed between the cytosol and membranes (Fig 1D). Interestingly, TRBP was exclusively cytoplasmic in primary astrocytes, while the majority of PACT and Dicer were associated with membranes (Fig 1E, Appendix Fig S1C). TRBP, PACT, and Dicer were largely excluded from membrane fractions in Hek293T cells (Fig 1F, Appendix Fig S1D), suggesting that the presence of membrane-associated RLCs is cell type-specific. To confirm that the RLC associates at neuronal membranes, we performed co-immunoprecipitation (coIP) experiments in membrane fractions of developing cortical neurons, isolated using sequential detergent extraction. Indeed, membrane-associated Dicer and Ago2 specifically co-precipitated with anti-TRBP antibody, used at two different concentrations (Fig EV1D). In summary, we show that the majority of the pre-miRNA processing complex is present at intracellular neuronal membranes, and in particular at the perinuclear and dendritic ER.

### BDNF stimulation induces the calcium-dependent redistribution of TRBP from the ER into the cytoplasm

We next investigated the dynamics of the RLC in developing cortical neurons during BDNF signaling. BDNF is secreted in response to neuronal activity and plays important roles in the activity-dependent development and plasticity of neuronal circuits [38,39]. Moreover, BDNF was previously shown to control the morphological plasticity of neurons via the regulation of several miRNAs [40–43]. BDNF led to a significant decrease in TRBP levels in ER microsomes isolated by differential centrifugation of cortical neurons, while the levels of other RLC components did not change significantly (Fig 2A). Importantly, the total levels of RLC proteins did not change significantly with BDNF application (Fig EV1E).

We further examined the subcellular distribution of TRBP in BDNF- or control-treated hippocampal neurons that were immunostained with TRBP and calnexin antibodies. We detected partial co-localization of TRBP and calnexin in both the soma and neuronal dendrites (Fig 2B). We quantified TRBP and calnexin co-localization

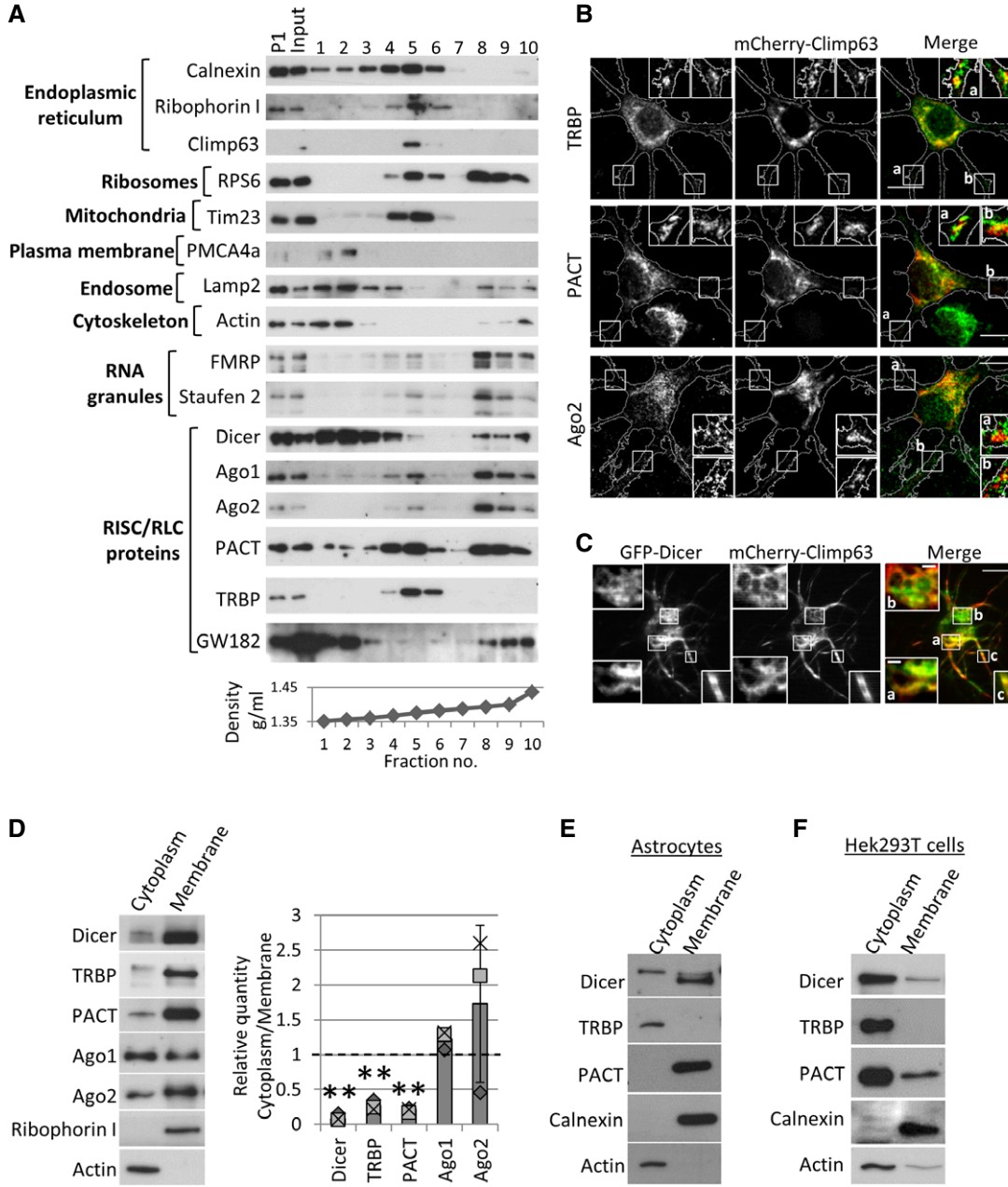

**Figure 1. The microRNA-generating complex localizes at the neuronal endoplasmic reticulum.**

A Components of the RLC co-precipitate with ER markers in density gradient fractionation. The postnuclear supernatant of cortical neurons was loaded underneath a continuous Optiprep density gradient. ER markers, calnexin, ribophorin I, and Climp63, co-sedimented with ribosomes (RPS6, *ribosomal protein subunit 6*), mitochondria (Tim23), and RLC proteins at fractions 3–5. The bottom graph shows the density of each fraction.

B TRBP, PACT, and Ago2 partially co-localize with the ER marker mCherry-CLIMP63. Hippocampal neurons (10 DIV) expressing mCherry-CLIMP63 were immunostained with TRBP, PACT, or Ago2 antibodies. Boxed insets are magnifications of primary dendrites as depicted by the adjacent letters (scale bars; 10 μm).

C GFP-Dicer partially co-localizes with mCherry-CLIMP63 at the neuronal soma and primary dendrites of hippocampal neurons. Neurons were imaged at 10 DIV using total internal reflection fluorescence (TIRF) microscopy using a penetration depth of 150 nm. Boxed insets are magnifications of primary dendrites as depicted by the adjacent letters (scale bars; 2 μm or 10 μm).

D The majority of Dicer, TRBP, and PACT are associated with neuronal membranes. Sequential detergent extraction was used to separate cytoplasmic from membrane fractions in 7 DIV cortical neurons. The relative quantity of depicted proteins was analyzed in three independent experiments (**$P < 0.005$, *t*-test type 3, $n = 3$, error bars s.d.). Actin and ribophorin I were used as loading controls for cytoplasmic and membrane fractions, respectively. The dotted line is placed at 1 to indicate the fold change. Individual data points acquired in three independent experiments are shown as separate shapes.

E TRBP is cytoplasmic in primary astrocytes. Sequential detergent extraction was performed in primary astrocyte cultures. Actin and calnexin were used as loading controls for cytoplasmic and membrane fractions, respectively.

F The majority of Dicer, TRBP, and PACT proteins are present in cytoplasmic fractions in HEK293T cells, isolated using sequential detergent extraction.

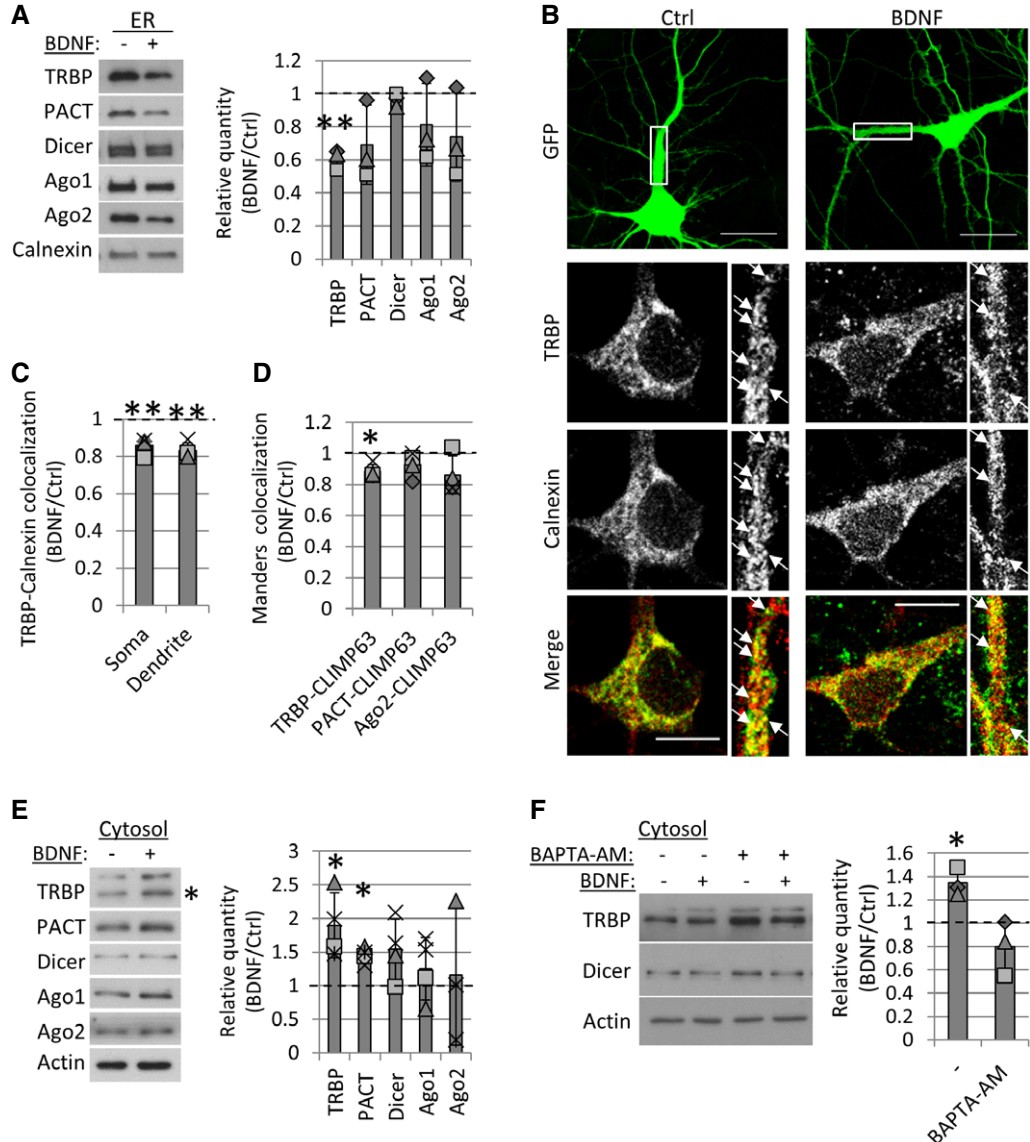

**Figure 2. Short BDNF stimulation leads to the redistribution of TRBP from the ER to the cytoplasm.**

A   BDNF stimulation leads to a decrease in TRBP levels from ER microsomes. Cortical neurons were stimulated BDNF with or vehicle control, and ER microsomes were isolated using differential centrifugation. Protein intensity values were normalized to calnexin in each of three independent experiments (**P = 0.008, t-test type 3, n = 3, error bars; s.d.).

B   Endogenous TRBP partially co-localizes with calnexin at the neuronal cell soma and primary dendrites. Hippocampal neurons were transfected with GFP at 7 DIV and fixed at 10 DIV following BDNF or control stimulation. Boxed insets are magnifications of the primary dendrite (15 μm in length). Arrows point to overlapping puncta (scale bars; 20 μm (GFP) or 10 μm).

C   BDNF leads to a decrease in co-localization between TRBP and calnexin at the soma and primary dendrites of hippocampal neurons. Manders' co-localization co-efficients between TRBP and calnexin were measured in the cell soma or primary dendrites (15 μm) using GFP as a morphological mask. Average co-localization values were normalized to controls in each experiment (**P < 0.008, t-test type 3, n = 4, error bars; s.d.).

D   BDNF decreases mCherry-CLIMP63 co-localization with TRBP, but not with PACT or Ago2. Hippocampal neurons were transfected with GFP and mCherry-CLIMP63 at 7 DIV and immunostained 3 days later, following BDNF or control stimulation. Manders' co-localization was measured at the cell soma using GFP as a mask as indicated (*P = 0.01, t-test type 3, n = 4, error bars; s.d.).

E   BDNF leads to an increase in the levels of TRBP (*P = 0.03) and PACT (*P = 0.004) in the cytoplasmic fraction. Cortical neurons (7 DIV) were control- or BDNF-stimulated, and cytoplasmic fractions were isolated using digitonin. Protein intensity values were normalized to actin in each experiment. The position of the star corresponds to the TRBP band used for analysis (~43 kDa) (t-test type 3, n = 4, error bars; s.d.).

F   The BDNF-induced increase in cytoplasmic TRBP is dependent on intracellular calcium. Cortical neurons (7 DIV) were treated with vehicle control, BDNF and BAPTA-AM as indicated, and cytoplasmic fractions were isolated using digitonin. Protein intensity values were normalized to actin in each independent experiment (*P = 0.03, n = 3, error bars; s.d.).

Data information: The dotted line is placed at 1 to indicate the fold change. Individual data points acquired in three independent experiments are shown as separate shapes.

using Manders' co-localization analysis and found that BDNF leads to a decrease in TRBP overlapping with calnexin at both the cell soma and dendrites of hippocampal neurons (Fig 2C). Similar results were obtained using Pearson's correlation analysis (Fig EV1F, Appendix Fig S2A). Moreover, BDNF caused a decrease in co-localization between TRBP and mCherry-Climp63, but not between PACT or Ago2 and mCherry-Climp63, at the soma of hippocampal neurons (Figs 2D, and EV1G, Appendix Fig S2B and C), thereby confirming our fractionation results (Fig 2A). We next isolated digitonin-permeabilized cytoplasmic fractions from BDNF- or control-treated cortical neurons and found that BDNF leads to an increase in cytoplasmic TRBP and PACT compared to controls (Fig 2E).

As it was previously suggested that BDNF leads to the phosphorylation of TRBP and increased binding to Dicer [43], we examined the role of phosphorylation in TRBP redistribution. We did not detect differences in binding to Dicer in co-IP or in the subcellular distribution of the reported phosphomimetic (S4D) and phosphonull (S4A) TRBP mutants compared to wild-type TRBP (Appendix Fig S3A and B). Furthermore, we did not find evidence of TRBP phosphorylation in neuronal cytoplasmic fractions treated with lambda phosphatase, in contrast to the heavily phosphorylated Myosin Va, which was used as a positive control (Appendix Fig S3C), suggesting that phosphorylation of TRBP does not mediate its redistribution to the cytosol. Instead, we found that the BDNF-induced increase in cytoplasmic TRBP was blocked by application of the intracellular calcium chelator BAPTA-AM (Fig 2F), while the total levels of Dicer and TRBP were unchanged (Fig EV1H), suggesting that TRBP redistribution to the cytoplasm is dependent on calcium. We were therefore prompted to further investigate the intracellular dynamics of RLC proteins as a possible mechanism to regulate RLC complexes and therefore pre-miRNA processing in neurons.

## BDNF leads to the dissociation of TRBP from Dicer and transiently inhibits pre-miR16 processing

We found that TRBP is dynamic upon BDNF stimulation, while Dicer remains stably associated with the ER (Fig 2A). We therefore speculated that BDNF might affect the interaction between TRBP and Dicer. Indeed, short BDNF stimulation of cortical neurons led to a robust decrease in the Dicer–TRBP interaction in co-IPs using two different TRBP antibodies as bait (Figs 3A and EV2A). This effect was not observed in co-IP from isolated cytoplasmic fractions (Fig EV2B), supporting the hypothesis that BDNF-dependent dissociation between TRBP and Dicer occurs at the membrane.

Several publications show that TRBP is important in Dicer-mediated cleavage of pre-miRNAs [28,44,45]. We therefore reasoned that the decrease in the TRBP–Dicer interaction following BDNF stimulation would lead to changes in pre-miRNA processing and therefore miRNA production. We profiled mature miRNA levels in neurons stimulated with BDNF or vehicle control for 20 min using small RNA sequencing. Overall, this brief stimulation protocol had relatively modest effects on mature miRNA levels (Figs 3B and EV2C). However, four miRNAs were differentially regulated in a highly reproducible manner ($P < 0.05$; Figs 3B and EV2D), suggesting that BDNF rapidly affects the biogenesis of a few neuronal miRNAs. Among them, miR-16-5p and miR-551b-5p levels were significantly reduced by BDNF (Figs 3B and EV2D), consistent with the reduced TRBP–Dicer interaction under these conditions (Fig 3A).

We further investigated the levels of miR-16-5p and miR-551b-5p as well as the levels of an unaffected neuronal miRNA, miR-138-5p, following 10 and 20 min of BDNF stimulation using real-time PCR (RT–PCR). We observed a significant decrease in relative levels of mature miR-16-5p after 10 min of BDNF stimulation, whereas the levels of miR-551b-5p and miR-138-5p did not change significantly (Fig 3C). Consistently, we observed a corresponding trend in the levels of pre-miR16, but not pre-miR551b or pre-miR138, which increased after 10 min of BDNF stimulation and returned back to basal at 20 min (Fig 3D). Moreover, pre-miR16 binding to TRBP was strongly reduced in BDNF-treated cortical neurons compared to control as assessed by anti-TRBP RNA IP (Fig 3E). Taken together, our results suggest that BDNF leads to decreased processing of pre-miR16 by TRBP-containing processing complexes, presumably as a result of TRBP dissociation from ER-associated Dicer.

## BDNF leads to changes in isomiR levels in cortical neurons

Although it is known that TRBP is an integral component of the RLC, the precise role of TRBP in pre-miRNA processing is controversial. For example, there are conflicting results on the role of TRBP in 5p versus 3p miRNA strand selection for loading onto Ago [45,46]. Consistent with the study by Kim *et al,* we did not observe any changes in the 5p to 3p ratio of miRNAs in small RNA sequencing following BDNF (Fig EV2E). It was further shown that TRBP affects the position of Dicer-mediated cleavage of pre-miRNAs, thereby promoting the formation of longer miRNA isomers, so called "isomiRs" [45,46]. Consistently, examination of our small RNA sequencing data shows that BDNF causes significant bi-directional changes in the relative proportion of 12 isomiRs annotated as canonical (Fig 4A, "0"), and 33 miRNAs with significant changes in relative proportion of at least one isomiR, following BDNF stimulation, including three members of the miR16 family (Fig EV3). Interestingly, the clustered miR16 family members, miR-16-5p and miR-15b-5p, had an increased proportion of short isomiRs at the expense of the longer, 22-nucleotide canonical isomiRs, following BDNF stimulation (Fig 4B). This is consistent with the exclusion of TRBP from Dicer-containing pre-miRNA processing complexes, which would favor the production of shorter isomiRs (Fig 4C). Thus, a BDNF-dependent isomiR switch may further contribute to the decrease in the total levels of miR-16-5p.

## MiR-16-5p mediates BDNF-induced dendritogenesis

It was previously shown that miR16 is important for neurogenesis *in vivo* [47]. However, little is known about its potential role in postnatal development. We first examined the regulation of miR-16-5p activity in young hippocampal neurons using a single-cell, dual-fluorescence sensor assay. We quantified endogenous miR-16-5p activity in hippocampal neurons transfected with a polycistronic vector expressing GFP and dsRed with two miR-16-5p binding sites within its 3′UTR ("miR16 sensor") or a control sensor (Fig 5A). There were higher numbers of dsRed-repressed cells in neurons transfected with the miR16 sensor compared to control, indicating the presence of endogenous miR-16-5p activity. Further, miR-16-5p activity was inhibited by the addition of an anti-sense locked nucleic acid (LNA) inhibitor ("LNA-miR16"), versus control LNA

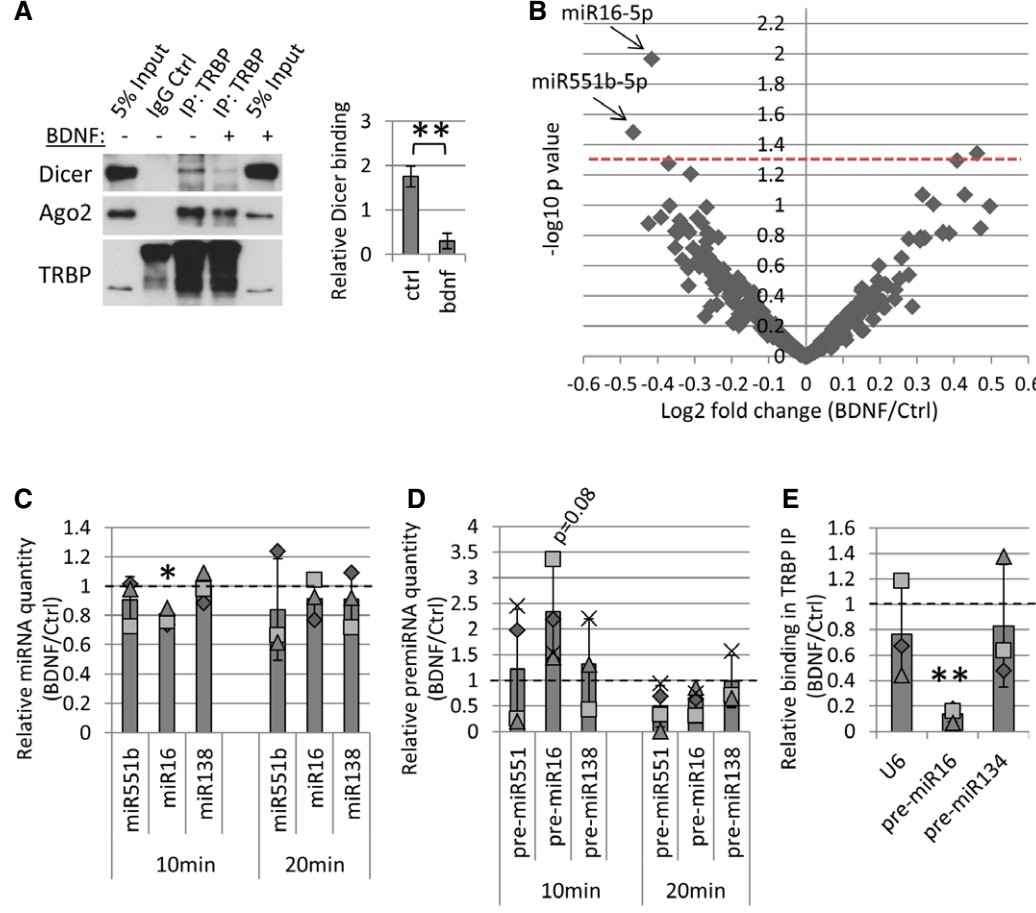

**Figure 3.   BDNF leads to TRBP dissociation from Dicer and decreased processing of pre-miR16.**

A    BDNF stimulation causes a decrease in the TRBP–Dicer interaction. Young cortical neurons were treated with BDNF or vehicle control, and co-immunoprecipitations (co-IPs) were performed using 2 μg control IgG or polyclonal anti-TRBP antibodies. Dicer binding in co-IP was normalized to inputs in each experiment (**$P$ = 0.001, $t$-test type 2, $n$ = 3, error bars represent s.d.).

B    Short BDNF stimulation changes the levels of only a few microRNAs, such as miR-16-5p and miR-551b-5p. Volcano plot of small RNA sequencing data from BDNF- or control-treated cortical neurons (6 DIV) was based on differential expression analysis using the edgeR package. Individual points represent the average fold change obtained in three independent experiments for each miRNA and plotted against obtained $P$-values. MiRNAs above the red dashed line are significant ($P$ < 0.05, $t$-test type 2, $n$ = 3).

C    BDNF stimulation leads to a transient decrease in mature miR-16-5p levels. 6 DIV cortical neurons were treated with BDNF for 10 or 20 min, and mature miRNA levels were measured in RT–PCR. $C_t$ values for each miRNA were normalized to U6 and to 20 min of control stimulation ($n$ = 3, *$P$ = 0.02, $t$-test type 3, error bars; s.d.).

D    BDNF stimulation leads to a transient increase in the levels of pre-miR16 after 10 min of BDNF stimulation. $C_t$ values obtained in RT–PCR were normalized to U6 and to 20-min control stimulation ($n$ = 4, $t$-test type 3, error bars; s.d.).

E    BDNF causes a decrease in pre-miR16 binding to TRBP. Cortical neurons were treated with BDNF or vehicle control, and RNA IPs were performed using a mouse monoclonal anti-TRBP antibody. $C_t$ values measured in RT–PCR were normalized to respective inputs ($n$ = 3, **$P$ = 0.002, $t$-test type 3, error bars represent s.d.).

Data information: The dotted line is placed at 1 to indicate the fold change. Individual data points acquired in three independent experiments are shown as separate shapes.

("LNA-Ctrl") (Fig 5B), demonstrating the specificity of this effect. Using this assay, we further show that endogenous miR-16-5p activity is reduced following 20 min of BDNF stimulation, but not after 24 h (Fig 5C), suggesting that the BDNF-induced block in pre-miR16 processing leads to a transient decrease in miR-16-5p activity.

We next examined the role of miR-16-5p in the maturation of hippocampal neurons using inhibition or overexpression of miR-16-5p. As expected, BDNF increased the dendritic complexity of young hippocampal neurons transfected with GFP and either LNA-Ctrl or an oligonucleotide control sequence for miRNA overexpression ("miR-Ctrl") (Figs 5D–G, and EV4A and C). Inhibition of miR-16-5p

activity using LNA-miR16 led to an increase in the dendritic complexity of hippocampal neurons at basal levels and occluded BDNF-induced dendritogenesis (Figs 5D and E, and EV4B). Conversely, overexpression of miR-16-5p blocked BDNF-induced dendritogenesis (Figs 5F and G, and EV4D). Further, overexpression of a Dicer binding-deficient truncation mutant of TRBP (GFP-TRBP Δ) led to an increase in dendrite complexity that occluded BDNF-induced dendritogenesis (Figs 5H and I, and EV4E and F), similarly to miR16 inhibition. Collectively, these results suggest that the transient, activity-dependent inhibition of miR-16-5p activity via the dissociation of TRBP and Dicer is crucial for BDNF-induced dendritogenesis.

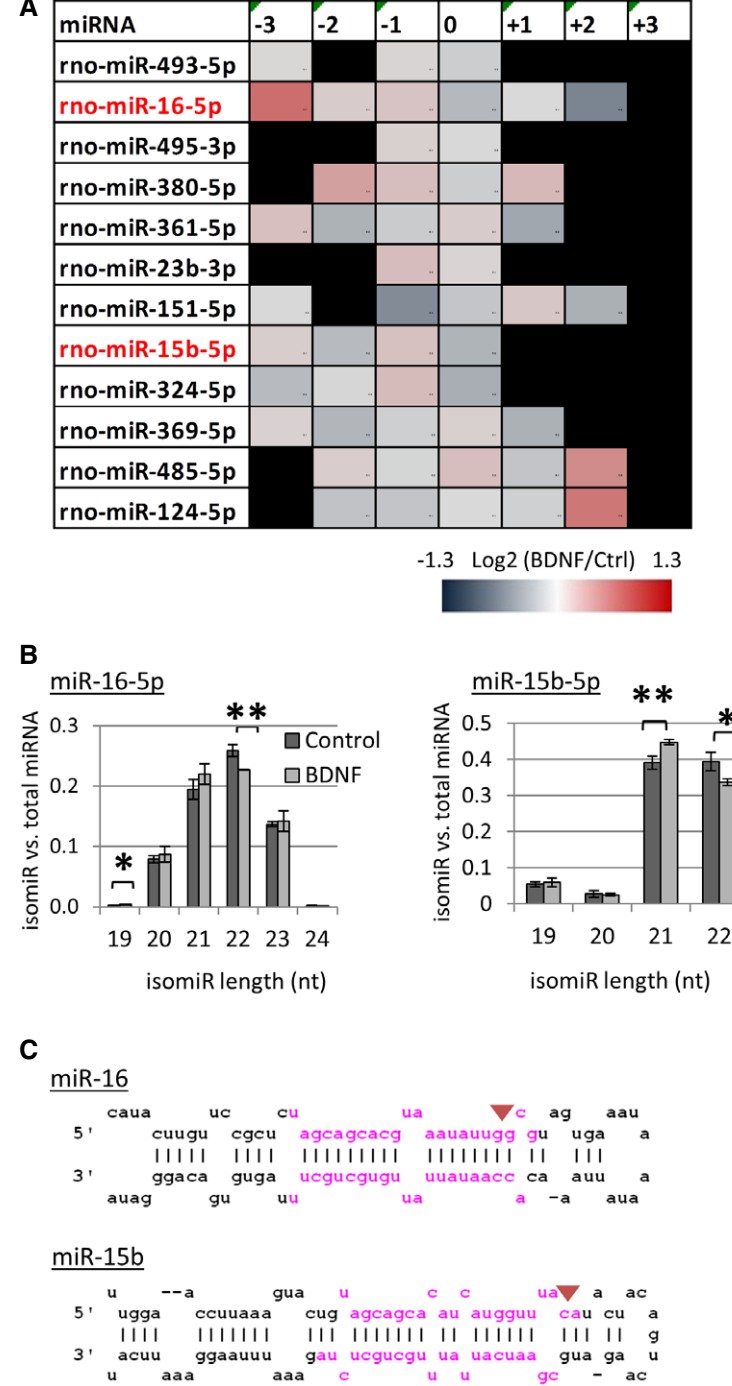

**Figure 4. BDNF changes the relative isomiRs levels for members of the miR16 family.**

A   BDNF changes the canonical isomiR proportion of a subset of miRNAs. IsomiR levels were compared to the total amount of the corresponding miRNA. Shown are the miRNAs that had significant changes in the canonical isomiR (annotated as "0", $P < 0.05$, t-test type 2, $n = 3$). Results are shown in order of decreasing significance. Shown in red are miRNAs of the miR16 family, miR-16-5p, and miR-15b-5p. Black squares correspond non-existing isomiRs. Numbers on the top of the table represent the trimming (negative) or addition (positive) of nucleotides to the 3′- or 5′-end of the canonical 5p or 3p isomiRs, respectively. Only templated nucleotide additions and trimmings, which might reflect Dicer-mediated pre-miRNA cleavage, were considered for analysis.

B   BDNF stimulation leads to a decrease in the canonical (22 nucleotides) isomiRs of the miR16 family, miR-16-5p (**$P = 0.005$), and miR-15b-5p (*$P = 0.02$). In both cases, this is accompanied by an increase in the levels of shorter isomiRs; the 19-nucleotide isoform of miR-16-5p (*$P = 0.04$); and the 21-nucleotide isoform of miR-15b-5p (**$P = 0.007$). The relative isomiRs proportion was calculated as described in A (t-test type 2, $n = 3$, error bars; s.d.).

C   Schematic representation of the pre-miRNA sequence and stem-loop structure for pre-miR16 and pre-miR15b. The 5p and 3p canonical miRNA sequences are highlighted in pink. The red arrows depict the predicted Dicer cleavage sites that would lead to shorter isomiRs induced by BDNF, as shown in (B).

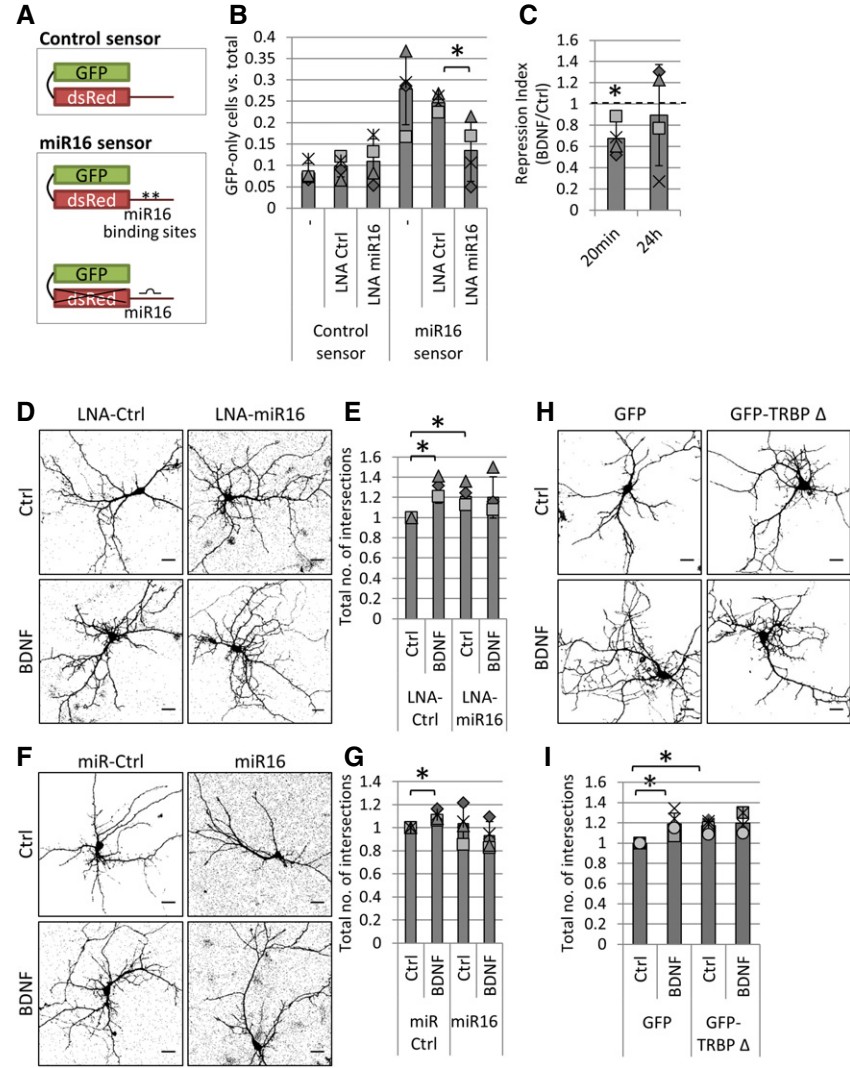

**Figure 5.  MiR-16-5p mediates BDNF-induced dendritogenesis of hippocampal neurons.**

A    Schematic of single-cell dual-fluorescence miRNA sensor assay. The control sensor plasmid consists of GFP and dsRed expressed under a separate promoter (top panel). The miR16 sensor includes two miR-16-5p binding sites at the 3′UTR of dsRed. Binding of endogenous miR-16-5p to the dsRed mRNA would inhibit the translation of dsRed but not GFP (bottom panel).

B    Endogenous miR-16-5p activity in developing hippocampal neurons. Hippocampal neurons were transfected with 500 ng miR16 or control sensor at 4 DIV and with or without LNA control ("LNA-Ctrl") or the anti-sense miR-16-5p inhibitor ("LNA-miR16"). Cells were fixed at 7 DIV, and the number of neurons expressing GFP and not dsRed ("GFP-only") was compared to the total number of transfected neurons. LNA-miR16 reduces the amount of repression of the miR16 sensor compared to LNA-Ctrl ($n$ = 4, *$P$ = 0.02, $t$-test type 2, error bars represent s.d.).

C    Short-term BDNF stimulation inhibits endogenous miR-16-5p activity. Hippocampal neurons were transfected at 4 DIV and treated with BDNF either 24 h or 20 min before fixation, and neurons were counted as in (B). Repression index was obtained by normalizing the miR16 sensor to the control sensor condition in each experiment ($n$ = 4, *$P$ = 0.02, $t$-test type 3, error bars; s.d.).

D, E   Inhibition of miR-16-5p increases dendritic complexity and occludes BDNF-induced dendritogenesis. (D) Hippocampal neurons were transfected with the control sensor (A) and stimulated with control or BDNF 24 h before fixation. Pictures were obtained on a confocal microscope, and thresholds were set to dsRed signal fluorescence in order to distinguish individual dendrites (scale bars are 10 μm). (E) To quantify dendritic complexity, the total number of dendritic intersections was calculated using Sholl analysis. Values were normalized to LNA-Ctrl and control treatment ($n$ = 3, *$P$ < 0.04, $t$-test type 3, error bars represent s.d.).

F, G   Overexpression of miR-16-5p blocks BDNF-induced dendritogenesis. (F) Hippocampal neurons were transfected with control sensor and either control miRNA mimic (miR-Ctrl) or a miR-16-5p mimic (miR16) and treated 2 days later with BDNF or control vehicle. DsRed signal fluorescence was used to set a threshold on images obtained on a confocal microscope (scale bars; 10 μm). (G) Total number of intersections was quantified in Sholl analysis, and values were normalized to miR-Ctrl, control treatment in each experiment ($n$ = 4, *$P$ = 0.02, $t$-test type 3, error bars; s.d.).

H, I   Overexpression of the Dicer binding-deficient deletion mutant of TRBP (GFP-TRBP Δ) occludes BDNF-induced dendritogenesis. (H) Hippocampal neurons were transfected with 200 ng dsRed and either 50 ng GFP or 200 ng GFP-TRBP Δ, and treated 2 days later with BDNF or control vehicle. DsRed signal fluorescence was used to set a threshold on images obtained on a confocal microscope (scale bars; 10 μm). (I) Total number of intersections was quantified in Sholl analysis, and values were normalized to control treatment in each experiment ($n$ = 5, *$P$ < 0.02, $t$-test type 3, error bars; s.d.).

Data information: The dotted line is placed at 1 to indicate the fold change. Individual data points acquired in three independent experiments are shown as separate shapes.

### BDNF stimulation up-regulates BDNF mRNA translation via miR-16-5p

We were interested in identifying target mRNAs that may mediate the effect of miR-16-5p on BDNF-induced dendritogenesis. Interestingly, the BDNF mRNA itself has a conserved target site for the miR16 family of miRNAs within its short 3′UTR (Fig 6A). Furthermore, miR-16-5p levels increase in an animal model of depression, which is inversely correlated with decreased levels of BDNF mRNA *in vivo* [48]. We therefore tested whether the BDNF 3′UTR is a direct target of miR-16-5p in primary cortical neurons. We transfected developing cortical neurons with a dual-luciferase reporter consisting of the short 3′UTR of the BDNF mRNA ("3′UTR WT") or a miR-16-5p-binding site mutant ("3′UTR Mut"), and either miR-Ctrl or miR-16-5p. Overexpression of miR-16-5p reduced the expression of 3′UTR WT but not 3′UTR Mut (Fig 6B), suggesting that the miR-16-5p binding site within the BDNF 3′UTR is indeed functional. Furthermore, BDNF transiently increased the expression of 3′UTR WT but not 3′UTR Mut (Appendix Fig S4), which was blocked by miR-16-5p overexpression (Fig 6C). Given the growth-promoting effect of BDNF, this result is consistent with the observed role of miR-16-5p as an inhibitor of dendritic outgrowth. In compartmentalized hippocampal neuron cultures, we could robustly detect pre-miR16 in neuronal processes by qPCR, to a similar extent as the known dendritically enriched pre-miR341 and in contrast to the somatically enriched pre-miR137 (Fig 6D) [20]. This is consistent with local pre-miR16 processing and local regulation of translation in the dendritic compartment.

### Expression of a membrane-targeted TRBP rescues the BDNF-induced decrease in miR-16-5p and blocks BDNF-induced dendritogenesis

Since the intracellular localization of TRBP is dynamic upon BDNF, we examined whether the relocalization of TRBP to the cytosol is important for the BDNF-induced decrease in miR-16-5p activity. We generated a membrane-targeted GFP-TRBP ("mGFP-TRBP") by adding a palmitoylation signal at the N-terminus of GFP. We reasoned that mGFP-TRBP would be associated with membranes even after BDNF stimulation and could therefore be utilized to examine the importance of TRBP localization in miR-16-5p production during BDNF signaling. Using sequential detergent extraction in Hek293T cells transfected with GFP, GFP-TRBP, or mGFP-TRBP, we validated that the majority of mGFP-TRBP is present in the membrane fraction, whereas most but not all of GFP-TRBP is present in the cytoplasm (Fig EV5A). In addition, mGFP-TRBP had perinuclear localization whereas GFP-TRBP displayed a more diffuse distribution pattern in the cell soma of hippocampal neurons (Fig EV5B). Moreover, Dicer associated with mGFP-TRBP more strongly compared to GFP-TRBP in co-IP experiments in Hek293T cells (Fig 7A), which is consistent with the hypothesis that intracellular membranes favor the association of Dicer and TRBP complexes.

Next, we transfected developing hippocampal neurons with the dual-fluorescence miR16 sensor (Fig 5A) and either GFP, GFP-TRBP, or mGFP-TRBP, and quantified miR-16-5p activity in neurons treated with BDNF or vehicle control. Consistent with previous experiments, BDNF reduced the miR-16-5p-mediated repression of dsRed in cells co-transfected with GFP (Fig 7B). Data from

mGFP-TRBP- but not GFP-TRBP-expressing neurons were significantly different from GFP control (Fig 7B). At the functional level, expression of mGFP-TRBP blocked BDNF-induced dendritogenesis (Figs 7C and D, and EV5C), similarly to miR-16-5p overexpression (Figs 5F and G, and EV4C and D). Consistent with the dual-fluorescence assay (Fig 7B), expression of GFP-TRBP had a partial effect on BDNF-induced dendritogenesis that is not significantly different to the GFP control condition (Figs 7C and D, and EV5C). The observed association of a small fraction of GFP-TRBP to intracellular membranes might explain its partial effects in these assays (Fig EV5A and B). Collectively, these results suggest that the targeting of TRBP to neuronal membranes inhibits the BDNF-induced decrease in miR-16-5p activity that is necessary for dendritogenesis. Therefore, the dynamic redistribution of TRBP from intracellular membranes, concomitant to its dissociation from Dicer, inhibits miR-16-5p production and thus leads to the derepression of neuronal target mRNAs such as BDNF, which is necessary and sufficient for BDNF-dependent dendritogenesis.

## Discussion

### MiRNA-generating complexes at intracellular membranes

We show that the major proteins of the miRNA-generating complex co-purify with ER markers in developing neurons (Figs 1A and EV1A). Notably, a major pool of Dicer co-sediments at light density fractions corresponding to smooth ER, the plasma membrane and endosomes, that is devoid of TRBP (Fig 1A). As PACT was also present at these fractions (Fig 1A), and PACT and TRBP were shown to have differential roles in pre-miRNA processing *in vitro* [45,49], we speculate the presence of distinct pools of Dicer-containing complexes, with distinct miRNA production profiles. It is worth noting, however, that Dicer has miRNA-independent functions [50] that may also be represented at these sites.

As TRBP is almost exclusively found at the rER (Figs 1A and EV1A, Appendix Fig S1A), and Dicer and TRBP interact at neuronal membrane fractions (Fig EV1D), we propose that the rER forms a selective platform for TRBP-containing pre-miRNA processing complexes, which may be important for the local regulation of miRNA production and activity. For instance, TRBP may specifically regulate miRNAs whose target mRNAs are translated at the rER, consistent with our observation that miR-16-5p regulates BDNF (Fig 6A–C), a secretory protein that undergoes translation at the rER. Interestingly, it was previously shown that the rER is specifically localized at dendritic branch points, where it directs the secretion of newly formed proteins, such as AMPA receptors [51]. We detected co-localization of TRBP, PACT, and Ago2 with the rER marker mCherry-Climp63 in both perinuclear regions and at dendritic branch points of hippocampal neurons (Fig 1B, Appendix Fig S2B), suggesting that miRNA production and translational repression may also preferentially occur at these sites. Moreover, we found that pre-miR16 is found in both neuronal processes and the cell soma (Fig 6D), supporting the hypothesis that pre-miR16 processing occurs at the dendritic rER.

Interestingly, we find that Dicer and PACT, core components of the miRNA-generating complex, preferentially associate with membranes in primary neurons and astrocytes. This is in stark contrast to

**A**

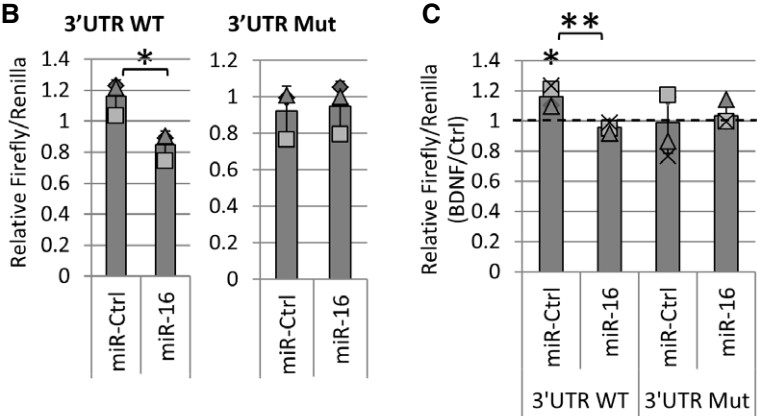

**B**  3′UTR WT    3′UTR Mut

**C**

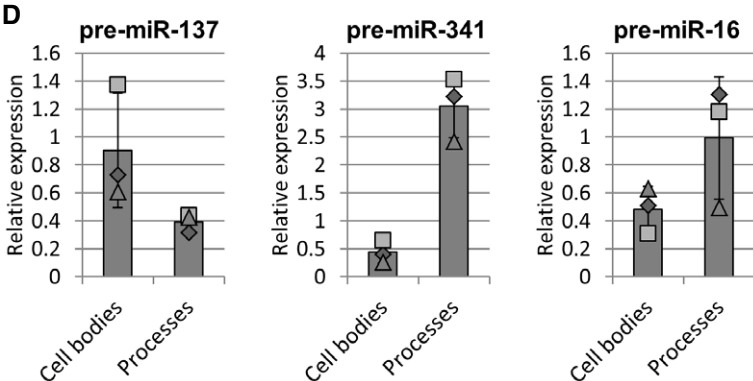

**D**  pre-miR-137    pre-miR-341    pre-miR-16

**Figure 6.  The BDNF mRNA is a miR-16-5p target and is regulated during BDNF stimulation.**

A  Diagram showing the region of the BDNF 3′UTR bound by miR16 family of miRNAs that is conserved across species. Nucleotides colored in red depict the binding site of miR-16-5p (sequences obtained from www.targetscan.org).

B  MiR-16-5p targets the 3′UTR of the BDNF mRNA. Cortical neurons were transfected with 100 ng pmiRGlo dual-luciferase reporter consisting of the 3′UTR of BDNF (3′UTR WT) or a miR-16-5p-binding site mutant (3′UTR Mut) and either miR-Ctrl or miR-16. The firefly-to-Renilla luciferase ratio was measured, and values were normalized to the respective reporter expression without miRNA mimic co-transfection ($n$ = 3, *$P$ = 0.02, $t$-test type 2).

C  BDNF stimulation leads to an increase in the translation of 3′UTR WT (*$P$ = 0.02, $t$-test type 3) but not 3′UTR Mut luciferase reporter, which is rescued by miR-16-5p overexpression (**$P$ = 0.002; $t$-test type 2). Cortical neurons were transfected as in (B), stimulated with BDNF or vehicle control for 20 min, and lysed 2 h after stimulation ($n$ = 4).

D  Pre-miR16 is present in the neuronal cell soma and processes of hippocampal neurons. Neurons were plated on compartmentalized chambers, and RNA was isolated from the neuronal cell body and processes separately. Pre-miR137 and pre-miR341 were used to assay somatic and neurite enrichment, respectively. Raw $C_t$ values were normalized to GAPDH mRNA in each experiment ($n$ = 3).

Data information: The dotted line is placed at 1 to indicate the fold change. Individual data points acquired in three independent experiments are shown as separate shapes.

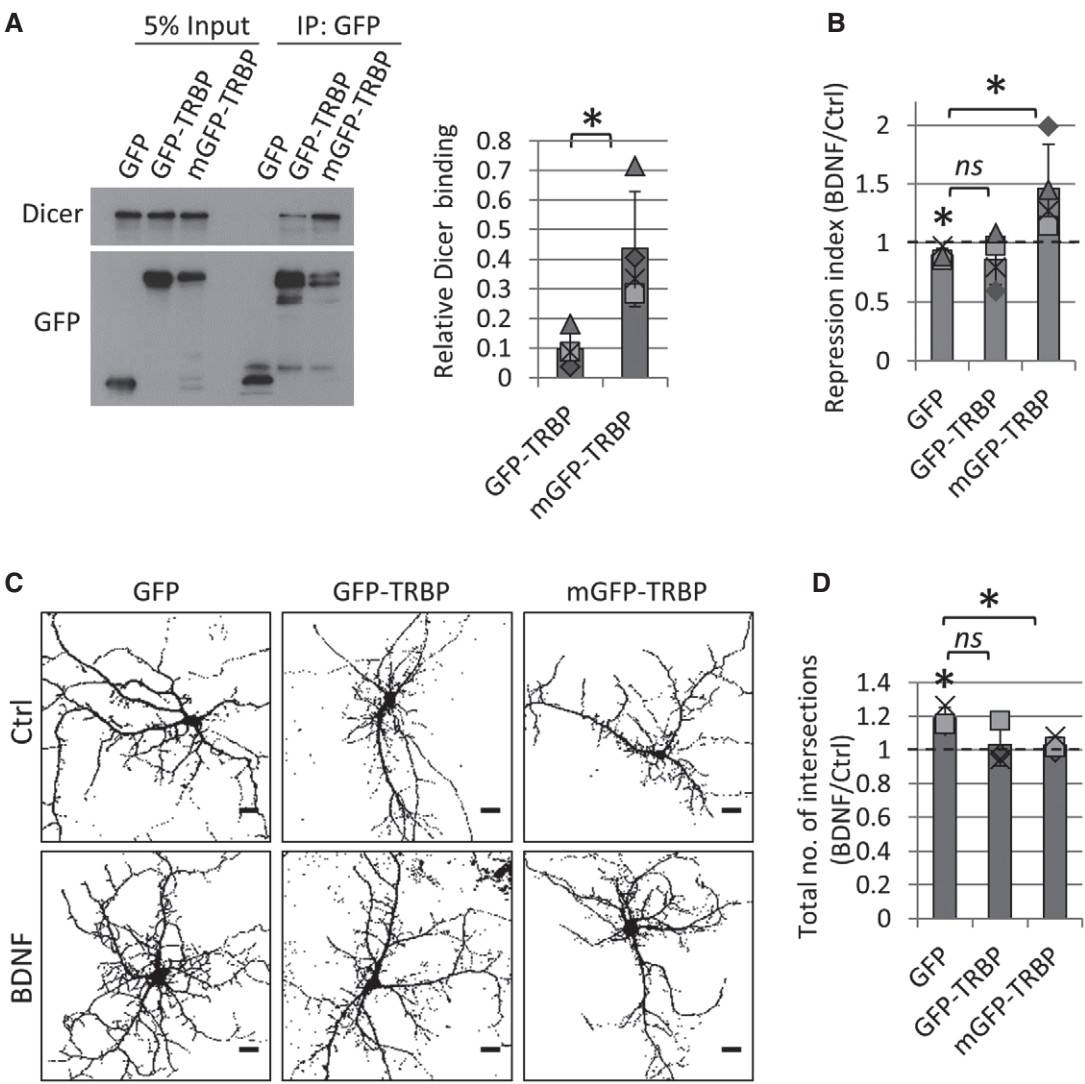

**Figure 7. Membrane-associated TRBP expression blocks the BDNF-induced decrease in miR16 and BDNF-induced dendritogenesis.**

A  More Dicer binds to membrane-targeted GFP-TRBP (mGFP-TRBP) compared to GFP-TRBP. HEK293T cells were transfected with GFP, GFP-TRBP, or mGFP-TRBP, and cell lysates were processed for co-IP using anti-GFP antibody. Dicer binding was normalized to respective input lanes in each condition (n = 4, *P = 0.02; t-test type 2, error bars are s.d.).

B  mGFP-TRBP but not GFP-TRBP blocks the BDNF-induced decrease in miR-16-5p activity in dual-fluorescence sensor assay. Hippocampal neurons were transfected with 200 ng miR16 or control sensor (see Fig 5A) plus 50 ng GFP, 400 ng GFP-TRBP, or 400 ng mGFP-TRBP. Neurons were treated with BDNF or vehicle control prior to fixation and immunostained with anti-GFP antibody. BDNF led to a decrease in miR-16-5p activity in GFP-transfected neurons (*P = 0.03, n = 4, t-test type 3), which was significantly different from mGFP-TRBP-expressing neurons (*P = 0.04, n = 4, t-test type 2, error bars are s.d.).

C, D  BDNF-induced dendritogenesis is blocked by mGFP-TRBP overexpression. (C) Hippocampal neurons (4 DIV) were transfected for 3 days with 200 ng dsRed, and 50 ng GFP, 400 ng GFP-TRBP, or 400 ng mGFP-TRBP. Neurons were treated with BDNF or vehicle control 24 h before fixation. Pictures were obtained on a confocal microscope, and threshold was set for dsRed to distinguish individual dendrites (scale bars are 10 μm). (D) The total number of intersections was calculated in Sholl analysis, and values were normalized to control treatment in each of three independent experiments. BDNF led to a significant increase in dendritic complexity (*P = 0.04; t-test type 3), which was significantly different in cells transfected with mGFP-TRBP, but not GFP-TRBP (n = 3, *P = 0.03; t-test type 2, error bars; s.d.).

Data information: The dotted line is placed at 1 to indicate the fold change. Individual data points acquired in three independent experiments are shown as separate shapes.

Hek293T cells, where the large majority of these proteins are cytoplasmic (Fig 1E and F, Appendix Fig S1C and D). Intriguingly, we observed that in contrast to neurons, TRBP is mostly cytoplasmic in primary astrocytes (Fig 1E), suggesting that membrane TRBP may be particularly important for neuron-specific processes. Thus, the confinement of miRNA regulators at intracellular membranes may overcome the limitation of free diffusion and enable miRNA-mediated translational repression at specific cellular compartments. Supporting this notion, we find that the efficiency of Dicer and TRBP binding is significantly higher in Hek293T cells when TRBP is anchored to the membrane (Fig 7A). We propose that the membrane anchoring of miRNA-generating complexes is especially important in highly

differentiated cells, where the local regulation of translation is crucial for morphological and functional heterogeneity.

## The intracellular localization of TRBP is dynamic during BDNF stimulation

We show that the ER membrane serves as a platform for the dynamic regulation of TRBP during activity-dependent dendritic outgrowth. BDNF causes a decrease in ER-associated TRBP (Figs 2A–D, and EV1F and G, Appendix Fig S2) and decreases its binding to Dicer (Figs 3A, and EV2A and B). We also show that Dicer is rather immobile during BDNF stimulation (Fig 2A and E), suggesting that TRBP does not affect Dicer localization in neurons. As BDNF led to a calcium-dependent increase in the cytoplasmic pool of TRBP (Fig 2E and F), and mGFP-TRBP binds more strongly to Dicer in co-IP (Fig 7A), our results suggest that the activity-dependent redistribution of TRBP from the ER to the cytosol mediates its dissociation from Dicer. Further, as the chelation of intracellular calcium increased cytoplasmic TRBP at basal conditions (Fig 2F), we speculate that a yet-unidentified calcium-sensing protein mediates the recruitment of TRBP to the ER membrane.

Previous publications by the Meffert group suggest that BDNF-induced phosphorylation of TRBP stabilizes Dicer and stimulates miRNA production in primary neurons [43,52]. We did not detect any changes in Dicer or TRBP levels upon BDNF (Fig EV1E), nor did we observe increased binding of Dicer to the reported phosphomimetic TRBP mutant (S4D) in co-IP (Appendix Fig S3A). Moreover, BDNF had modest effects on mature miRNA levels in small RNA sequencing (Figs 3B and EV2C). To our knowledge, we are the first to directly assay the Dicer–TRBP interaction in co-IP following short-term BDNF stimulation (Fig 3A, Appendix Fig S1E) and using phosphorylation mutants of TRBP (Appendix Fig S3A). Consistent with our results, a structural study has suggested that the previously reported phosphorylation sites of TRBP are unlikely to affect its binding to Dicer [46]. In fact, higher molecular weight bands of neuronal TRBP did not respond to treatment with lambda phosphatase (Appendix Fig S3C), suggesting the presence of other post-translational modifications that may regulate the subcellular distribution and function of TRBP. As membrane-localized TRBP was particularly pronounced in neuronal cells, perhaps some of the discrepancies between our results and studies by the Meffert group may be due to the presence of non-neuronal cells in primary neuronal cultures. In fact, the use of anti-mitotics to eliminate proliferating cells has not been reported in the aforementioned studies.

## BDNF alters pre-miRNA processing

Our small RNA sequencing data revealed modest changes in the total levels of miRNAs and no change in strand selection upon BDNF (Figs 3B, and EV2C and E). Since TRBP dissociates from Dicer within the same time of BDNF stimulation (Figs 3A and EV2A), our results are consistent with a previous study that generated TRBP- and PACT-knockout HeLa cells and showed that neither TRBP nor PACT affects the abundance of miRNAs, Ago loading, or strand selection [45]. Rather, TRBP affects the fidelity of Dicer cleavage, thereby promoting the generation of longer isomiRs [45,46]. Consistently, BDNF stimulation changed the isomiR proportion for

33 miRNAs in young cortical neurons (Fig EV3), 16 of which were previously shown to co-purify at mammalian polyribosomes in neurons, including miR-16-5p and miR-15b-5p [36]. Furthermore, we detected significant changes in the canonical isomiRs of 12 miRNAs. Consistent with the proposed role of TRBP in promoting the production of longer isomiRs [45,46], we often observed an increase in the proportion of shorter isomiRs when the proportion of canonical isomiRs was decreased and *vice versa* (Fig 4A–C).

From our isomiR candidates, we could validate the BDNF-induced decrease in the total levels of miR-16-5p (Fig 3C), which was the most significantly down-regulated miRNA in our small RNA sequencing data (Fig 3B). Our results further suggest that this is due to decreased processing of pre-miR16 by TRBP-containing complexes (Fig 3C–E). As miR-16-5p was previously shown to target AU-rich element (ARE)-containing mRNAs [53], we tested for a potential enrichment in isomiRs containing the *UAAAU* sequence motif. However, we did not detect particular enrichment of AU-rich sequences in BDNF-affected isomiRs (approximately a third of isomiRs shown in Fig EV3).

Although the functional implications of isomiRs are so far unclear, it was previously shown that they can be incorporated into Ago complexes [54,55], and may affect Ago loading and miRNA stability [56]. Moreover, some 5′ isomiRs were shown to have different targets than canonical miRNAs, due to a shift in the seed region [57,58]. We detected two miRNAs, miR-9a-3p and miR-22-3p, with significant, BDNF-induced changes in their 5′ isomiRs; "+1" and "−1", respectively (Fig EV3). Functional annotation of isomiR targets provides preliminary evidence for a decrease in the repression of endosomal trafficking-related proteins upon BDNF (Appendix Table S1). This may reflect the requirement for the vast membrane expansion of dendritic trees during neuronal growth [59]. Interestingly, miR-22-3p, miR-9a-3p, and miR-124-5p (which was also affected by BDNF) were previously shown to regulate of neuronal growth and differentiation via the REST/CoREST transcriptional complex [60,61], which itself represses the expression of BDNF [62]. Thus, BDNF-induced isomiR changes may provide an additional layer of regulation of gene expression during neuronal development.

## BDNF-induced inhibition of miR-16-5p activity is important for dendritogenesis

We showed that the BDNF-induced decrease in pre-miR16 processing leads to a transient inhibition in miR-16-5p activity (Fig 5A–C), which was rescued by the expression of the membrane-targeted mGFP-TRBP (Fig 7B). Furthermore, we found that either inhibition of miR-16-5p or overexpression of a Dicer binding-deficient TRBP mutant led to an increase in the dendritic complexity of young hippocampal neurons, which was not further increased by BDNF (Figs 5D, E, H, I, and EV4A, B, E, F). On the other hand, overexpression of either miR-16-5p or mGFP-TRBP blocked BDNF-induced dendritogenesis (Figs 5F and G, 7C and D, EV4C and D, and EV5C). We therefore propose that the BDNF-induced redistribution of TRBP from intracellular membranes and the subsequent inhibition of miR-16-5p production and activity mediate BDNF-induced dendritogenesis in developing neurons. Indeed, we found that miR-16-5p targets the BDNF mRNA itself (Fig 6A and B) and that BDNF stimulation leads to the transient

increase in the translation of a BDNF 3′UTR luciferase reporter via miR-16-5p (Fig 6C, Appendix Fig S4). As BDNF is crucial for neuronal morphogenesis during development [38], targeting of the BDNF mRNA by miR-16-5p may underlie the inhibitory effect of miR-16-5p in dendritic complexity and its role in BDNF-induced dendritogenesis. Interestingly, it was previously shown that canonical Wnt signaling inhibits miR-16-5p and miR-15-5p production via post-transcriptional mechanisms [63] and that Wnt mediates expression of BDNF [64]. Furthermore, work by the Kellerman group has shown that miR-16 antagonizes Wnt signaling by targeting the serotonin transporter in hippocampal neurons [47,65]. Therefore, miR-16-5p may be part of an intricate cross talk between Wnt and BDNF signaling pathways that mediates neurogenesis and neuronal outgrowth.

In conclusion, we report a novel mechanism for the BDNF-induced regulation of pre-miRNA processing via the dynamic association of TRBP with ER membranes, which is necessary for BDNF-induced dendritogenesis. We speculate that the confined inhibition of miR-16-5p production and activity is important for an early signaling response to locally modulate the morphology and composition of neuronal dendrites, which is in turn crucial for physiological neuronal development.

## Materials and Methods

### Plasmids and oligos

mCherry-Climp63 was a gift from C. Hanus (French Institute of Health and Medical Research, Paris, France) [51].

For GFP-Dicer, the Dicer CDS was amplified from pFRT/TO/FLAG/HA-DEST DICER (a gift from Thomas Tuschl; Addgene #19881) [66] using *NheI*- and *AgeI*-flanked primers and inserted into peGFP-N1.

Fluorescent fusion constructs of TRBP and PACT were made using pcDNA-TRBP and pcDNA-PACT as templates (a gift from Dong-Yan Jin; Addgene plasmids #15666 and #15667) [67]. GFP was subcloned into the pcDNA3 vectors using *NheI* and *KpnI*. Membrane-tagged GFP-TRBP was made by insertion of *NheI–AgeI*-flanking oligonucleotides consisting of two palmitoylation sites from GAP-43. GFP-TRBP was used as a template to make TRBP phospho-null (S4A) and phospho-mimic (S4D) mutants and the Dicer-binding deletion mutant (GFP-TRBP Δ). Mutagenesis was performed using the Phusion site-directed mutagenesis kit (Thermo Scientific).

Bi-cistronic reporter constructs in single-cell fluorescent sensor assay (pTracer-CMV-dsRED) were previously described [68]. A *NotI*- and *XbaI*-flanked oligonucleotide containing two perfect binding sites for miR16 was subcloned into the 3′UTR of dsRed.

Pre-miR miRNA precursor-negative control (AM17110) and hsa-miR-16-5p (Assay ID: PM10339) from Ambion were used for control or miR16 overexpression. Anti-sense miRNA inhibitor control (199006-101) and rno-miR-16-5p inhibitor (4102067-100) were purchased from Exiqon. For detection of mature miRNAs, hsa-miR16-5p (Assay ID: 000391), rno-miR-551B-3p (Assay ID: 002760), hsa-miR-138-5p (Assay ID: 002284), mmu-miR-134-5p (Assay ID: 001186), and TaqMan U6 snRNA (Assay ID: 001973) were purchased from Life Technologies.

### List of primer sequences

| Name | Forward primer (5′–3′) | Reverse primer (5′–3′) |
|---|---|---|
| GFP-Dicer | TATGCTAGCGTCGACATGAAAAGCC | ATAACCGGTTGGCTATTGGGAACCTG |
| mGFP-TRBP | CTAGCATGCTGTGCTGTATGAGAAGAACCAAACAG; GTTGAAAAGAATGATGAGGACCAAAAGATCGCA | CTTTTCAACCTGTTTGGTTCTTCTCATACAGCACAGCATG; CCGGTGCGATCTTTTGGTCCTCATCATT |
| GFP-TRBP S4A | | |
| S142A/S152A | CTGCAGCCCCCTGTCGCCCCTCAGCAG | TTCCATGGGGGGGGGCCCTGGTTAGG |
| S283A | TCCGCAGTTGCGCCCTGGGCTCC | GGGACAGGATCTTCTCTCCTACTGAATTTCGTAGAGAATC |
| S283A/S286A | CGCCCTGGGCGCCCTGGGTGC | CAACTGCGGGAGGGACAGGATCTTCTCTCCTACTGAATTTC |
| GFP-TRBP S4D | | |
| S142D/S152D | CTGCAGCCCCCTGTCGACCCTCAGCAG | TTCCATGGGGGGGGTCCCTGGTTAGG |
| S283D | TCCGCAGTTGCGACCTGGGCTCC | GGGACAGGATCTTCTCTCCTACTGAATTTCGTAGAGAATC |
| S283D/S286D | CGACCTGGGCGACCTGGGTGC | CAACTGCGGGAGGGACAGGATCTTCTCTCCTACTGAATTTC |
| GFP-TRBP Δ | TGACTCGAGTCTAGAGGGCCCGTTTA | CGTGTGCACTCGAAGCAGCATTTTGG |
| miR16 sensor | GGCCGCCGCCAATATTTACGTGCTG; CTAGGCGCGCCCGCCAATATTTACGTGCTGCTAT | GGGCGCGCCTAGCAGCACGTAAATATTGGCGGC; CTAGATAGCAGCACGTAAATATTGGC |
| Pre-miR16 | CACGTAAATATTGGCGTAGTGAAAT | TATCCCTGTCACACTAAAGCAGC |
| pre-miR-137 | GGTGACGGGTATTCTTGGGT | CGACTACGCGTATTCTTAAGCAAT |
| pre-miR-341 | CCGATCGCTCGGTCTGT | ACCGACCGACCGATCG |
| Pre-miR551b | GTGCTCTTGTGGCCCATGAA | TCGCCTTCCTGTTCTGTACC |
| Pre-miR134 | GGGTGTGTGACTGGTTGACCA | GGGTTGGTGACTAGGTGGCC |
| Pre-miR138 | TGCAGCTGGTGTTGTGAATCA | GTGAAATAGCCGGGTAAGAGGAT |
| U6 snRNA | CTCGCTTCGGCAGCACA | AACGCTTCACGAATTTGCGT |
| GAPDH mRNA | GCCTTCTCTTGTGACAAAGTGGA | CCGTGGGTAGAGTCATACTGGAA |

### List of primary antibodies

| Name | Source; Catalogue number | Application | Dilution |
|---|---|---|---|
| TRBP | Abcam; ab129325 | IF | 1:100 |
| | Abcam; ab72110 | IB | 1:5,000 |
| PACT | Proteintech; 10771-1-AP | IF | 1:200 |
| | Abcam; ab75749 | IB | 1:1,000 |

| Name | Source; Catalogue number | Application | Dilution |
|---|---|---|---|
| Ago2 | Abcam; ab32381 | IF | 1:200 |
| | Rat serum, G Meister (University of Regensburg) | IB | 1:10 |
| Ago1 | Millipore; 04-083 | IB | 1:1,000 |
| Calnexin | Millipore; ab2301 | IF, IB | 1:1,000 |
| Ribophorin I | Santa Cruz; sc-12164 | IB | 1:200 |
| Climp63 | Abcam; ab84712 | IB | 1:500 |
| S6 ribosomal protein | Cell Signaling; 2317S | IB | 1:1,000 |
| Phospho-TrkA | Cell Signaling; 9141 | IB | 1:1,000 |
| HA | Abcam; ab9110 | IF | 1:400 |
| Dicer | Sigma; SAB4200087 | IB | 1:5,000 |
| GW182 | Sigma; SAB2102506 | IB | 1:1,000 |
| Lamp2 | Novus Biologicals; NB300-591 | IB | 1:1,000 |
| PMCA4 | Abcam; ab2783 | IB | 1:1,000 |
| Beta-actin | Sigma; A5441 | IB | 1:20,000 |
| FMRP | Abcam; ab17722 | IB | 1:5,000 |
| Cyclophilin A | Abcam; ab41684 | IB | 1:1,000 |
| GFP | Abcam; ab13970 | IF, IB | 1:2,000 |
| Staufen 2 | M. Kiebler (University of Munich) | IB | 1:5,000 |
| Tim23 | BD Biosciences; 611223 | IB | 1:1,000 |
| Myosin Va | Sigma, M4812 | IB | 1:1,000 |

IF, Immunofluorescence; IB, Immunoblotting.

## Cell culture

### Primary neuronal cell culture
Primary cortical and hippocampal cultures were prepared from embryonic day 18 (E18) male and female Sprague Dawley rats (Charles River Laboratories, Sulzfeld, Germany) as previously described [40]. Dissociated hippocampal neurons and cortical neurons were seeded on poly-L-lysine-/laminin-coated coverslips and plates coated with poly-L-ornithine, respectively. Neurons were maintained in Neurobasal media containing 2% B27, 2 mM GlutaMax, and 100 μg/ml streptomycin, and 100 U/ml penicillin (Gibco, Invitrogen) in a humidified incubator with 5% $CO_2$ at 37°C. Cortical neuronal cultures were treated with fluorodeoxyuridine (FUDR; Sigma; F-0503) + uridine (Sigma; F-303) (final concentration 10 μM) from 4 DIV to stop proliferation of non-neuronal cells.

Neurons were transfected using Lipofectamine 2000 (Invitrogen). Primary neurons were seeded in 24-well plates, and 1 μg DNA was transfected per well. An empty pcDNA3 vector was used to make up the total amount of DNA for transfection. Neurons were transfected in Neurobasal medium containing 2% B27 and 2 mM GlutaMax for 1 h, which was washed out and replaced with conditioning medium. Transfected cells were processed in further experiments 48 to 72 h later.

For compartmentalized neuron cultures, dissociated hippocampal neurons (19 DIV) were plated onto 1-mm pore and 30-mm-diameter polyethylene terephthalate (PET) membrane filter inserts (Millipore) as described previously [20] .

Primary mouse astrocytes were kindly provided by Prof. C. Culmsee (University of Marburg).

### HEK293T cell culture
HEK293T cells (Sigma) were maintained in DMEM media plus 10% fetal bovine serum (FBS), 1 mM glutamine, 100 U/ml penicillin, and 100 μg/ml streptomycin (Invitrogen), in a humidified incubator with 5% $CO_2$ at 37°C. HEK293T cells were transfected using the calcium phosphate method with a final $CaCl_2$ concentration of 0.1 M. Cells were washed 5 h after transfection and processed in coIP 24–48 h later.

### Treatments
Stock solutions of hBDNF (Peprotech, 450-02) were made in sterile water and 0.1% BSA, as per manufacturer's instructions. Unless otherwise stated, 6–7 DIV neurons were stimulated with BDNF (100 ng/ml) or vehicle control (0.1% BSA) for 20 min. BAPTA-AM (15551, Bertin Pharma) was dissolved in DMSO and used at the concentration of 10 μM for a total of 30 min.

## Microscopy

### Immunofluorescence
Hippocampal neurons were fixed in 4% paraformaldehyde/sucrose at room temperature for 20 min. Cells were permeabilized in blocking buffer consisting of 5% normal goat serum (NGS) and 0.5% saponin in phosphate-buffered saline (PBS). Primary antibodies were incubated in blocking buffer at corresponding dilutions (see List of primary antibodies). Alexa-488-, Alexa-546-, and Alexa-647-conjugated secondary antibodies (Invitrogen) were diluted in blocking buffer at 1:2,000 and incubated at room temperature for 45 min. Mitotracker Deep Red (M22426, Invitrogen) was a kind gift from M Buenemann (BPC, Marburg, Germany).

For single-cell dual-fluorescence sensory assay, transfected hippocampal neurons were fixed and immunostained with anti-GFP (1:1,000; ab13970 Abcam) for 1 h, and anti-chicken Alexa-488-conjugated secondary antibody for 45 min at room temperature, both diluted in 0.02% gelatin/0.5% Triton X-100/PBS.

### Co-localization analysis
Images were acquisitioned on Leica SP5 using 63× objective (1,024 × 1,024 pixel). Image processing and analysis were performed on Fiji software [69]. The Coloc 2 algorithm was used to obtain co-localization values with automatic threshold. The average co-localization of approximately ten cells was measured in each condition, in each of three or four experiments. Experiments were performed in a blinded manner.

### Single-cell dual-fluorescence sensor assay
To quantify miRNA-mediated repression, neurons were counted using a 20× objective and the number of neurons expressing GFP but not dsRed was compared to the total number of transfected neurons. Approximately 100 neurons were counted per experimental condition, in three or four independent experiments. Experiments were performed in a blinded manner.

### Dendritogenesis assay

XY scans (×20 objective, 1,024 × 1,024 pixels) were taken on a confocal laser-scanning microscope (LSM 5 Pascal; Zeiss, Germany). A grid of concentric circles spaced 15 μm apart was placed around the cell soma of GFP-expressing neurons, and the number of dendrites crossing each circle was counted manually on images thresholded in ImageJ. The average number of intersections was calculated for approximately 10 cells per each condition, in each of three or four independent experiments. All experiments were performed in a blinded manner.

### Total internal reflection fluorescence

Total internal reflection fluorescence imaging was performed on a Leica DMI 6000 B microscope equipped with a heated incubation chamber, using a 100× oil objective (Leica Microsystems). Live neurons were imaged at 37°C in HEPES-buffered saline (140 mM NaCl, 5 mM, KCl, 1.8 mM CaCl$_2$, 0.8 mM MgCl$_2$, 10 mM glucose, 25 mM HEPES, pH 7.5). The osmolarity of imaging medium was adjusted to match the osmolarity of the neurons at the day of imaging, using sucrose or water. Acquired images were processed using the Leica LAS AF software package.

### Subcellular fractionation

Unless otherwise stated, all steps were carried out at 4°C or on ice. ETDA-free complete protease inhibitors and PhosSTOP phosphatase inhibitors (Roche) were freshly added to all the buffers.

For density gradient centrifugation, cultured cortical neurons were treated with 200 mM cycloheximide (Sigma) for 5 min at 37°C before being washed three times with cold PBS and lifted from the cell culture plate in isotonic extraction buffer (IEB; 10 mM HEPES pH 7.8, 250 mM sucrose, 25 mM KCl, 1 mM EGTA, 2 mM MgCl$_2$, and 200 mM cycloheximide). Cells were pelleted at 800 × $g$ for 5 min and incubated in three times the packed cell volume (PCV) of hypotonic extraction buffer (HEB; 10 mM HEPES pH 7.8, 25 mM KCl, 1 mM EGTA, and 200 mM cycloheximide) for 20 min. Swollen cells were then pelleted and resuspended in two times the PCV with IEB including TURBO DNAse (2 μl/ml). Cells were homogenized with ten strokes in Dounce homogenizer followed by five times with a 27-gauge needle and left on ice for 15 min. Nuclei were pelleted by centrifugation 700 × $g$, 10 min, and the postnuclear supernatant (PNS) was mixed with 50% Optiprep density gradient medium (D1556, Sigma) to make a 35% solution and loaded underneath a 2.5–30% continuous Optiprep gradient. Gradients were centrifuged in a Beckman SW41 rotor at 48,000 × $g$ for 18 h, with slow acceleration and deceleration. Equal volume fractions were collected and processed in Western blotting. The density of the fractions was measured using a refractometer.

The ER isolation kit (RER0100, Sigma) was used to isolate ER microsomes, according to the manufacturer's protocol.

The protocol for sequential detergent extraction was adapted from Stephens et al [70]. Cultured cells were washed twice with ice-cold PBS and permeabilization buffer (110 mM KOAc, 2.5 mM Mg(OAc)$_2$, 1 mM EGTA, 25 mM K-HEPES pH 7.2, 1 mM dithiothreitol, and 0.5% digitonin) was added to the cells and the plate incubated for 5 min on ice with gentle agitation. The cytoplasmic fraction was collected, and the cells were gently washed with permeabilization buffer containing 0.004% digitonin. The membrane fraction was isolated using lysis buffer (400 mM KOAc, 15 mM Mg(OAc)$_2$, 25 mM K-HEPES pH 7.2, 1 mM dithiothreitol, 1% (v/v) Nonidet P-40, and 0.5% (w/v) deoxycholate), for 5 min with gentle agitation. Dithiothreitol and digitonin were added fresh. The cytosolic and membrane fractions were centrifuged at 7,500 × $g$ for 10 min before samples were processed further.

### Immunoprecipitation

Cultured cells were washed twice with ice-cold PBS and lysed on ice in buffer containing 0.5% Nonidet P-40 (Thermo), 300 mM NaCl, 2.5 mM MgCl$_2$, and 20 mM Tris–HCl pH 7.4. Complete EDTA-free protease and phosphatase inhibitors were added fresh. Cell lysates were kept on ice for 15–30 min prior to centrifugation at 15,000 × $g$ for 15 min at 4°C. Cleared lysates were incubated with 2–5 μg of anti-TRBP antibody (mouse monoclonal or rabbit polyclonal), anti-GFP (NBP2-43575; Novus Biologicals), or normal mouse or rabbit IgG (sc-2025 and sc-2027; Santa Cruz) for 1 h at 4°C with rotation. Dynabeads Protein G (10004D, Thermo) for mouse antibodies or Protein A-Sepharose beads (P7786, Sigma) for rabbit antibodies were then incubated for 1 h at 4°C with rotation, and beads were washed four times in lysis buffer and processed for Western blot or RT–qPCR. For RNA IPs, buffers were made in DEPC-treated water and contained SUPERase In RNAse Inhibitor (0.2 U/μl, AM2696, Invitrogen).

### Immunoblotting

Protein extracts were diluted in Laemmli sample buffer, separated in SDS–PAGE and blotted onto a polyvinylidene difluoride (PVDF) membrane. Non-specific antibody binding was blocked in Tris-buffered saline containing 5% milk powder or 5% BSA and 0.2% Tween-20. Primary antibodies were incubated in blocking buffer at the corresponding dilutions. Horseradish peroxidase (HRP)-conjugated secondary antibodies were as follows: goat anti-rabbit (1:10,000; 401315 Calbiochem), rabbit anti-mouse (1:10,000; 402335; Calbiochem), rabbit anti-goat (1:20,000; 401515; Calbiochem), goat anti-rat (1:10,000; 401416; Calbiochem), and rabbit anti-chicken (1:20,000; ab6753 Abcam).

### Quantitative real-time PCR

Total RNA was isolated using mirVana miRNA isolation kit (AM1561, Invitrogen) or peqGOLD Trifast (30-2010, VWR) for RNA immunoprecipitation. Genomic DNA contamination was eliminated with TURBO DNA-free kit (AM1907, Invitrogen). Reverse transcription of RNA was carried out using iScript cDNA synthesis kit (170-8891, Bio-Rad) or TaqMan RT–qPCR kit (4366597, Invitrogen), according to manufacturer's protocols. Relative cDNA quantification was performed using StepOnePlus Real-Time PCR system (Applied Biosystems) using iTaq SYBR Green Supermix with ROX (172-5121, Bio-Rad) for detection of precursor miRNAs and TaqMan Universal PCR Master Mix (4364341, Invitrogen) for detection of mature miRNAs.

### Small RNA sequencing

Six small RNA libraries, representing 3 biological replicates from control- or BDNF-treated cortical neurons, were prepared using

NEBNext® Multiplex Small RNA Library Prep Set for Illumina® (Set 1) kit (New England BioLabs) as per manufacturer's instructions. Multiplexed small RNA libraries were sequenced for 50 cycles in a single lane of one Illumina HiSeq2000 flow cell (EMBL Heidelberg). Raw sequencing reads were trimmed from 3′ adapter (AGATCG-GAAGAGCACACGTCT) and filtered according to quality using "percent = 90 and cutoff = 30" parameters of Fastx-Toolkit for fastq data on *Galaxy* [71] (https://usegalaxy.org/). Reads were excluded if they only contained the adapter sequence, did not contain the adapter sequence prior to trimming, or were shorter than 15 nucleotides. The remaining reads were mapped to the rat mature miRNAs (miRBase v21) using default parameters (one mismatch, 3 nt in the 3′ or 5′-trimming variants, 3 nt in the 3′-addition variants) of Mira-ligner software [72]. For mature miRNA analysis, only those that are represented by at least 10 reads per million (RPM) in one of the conditions were considered. Differential expression analysis of miRNAs was performed using *edgeR* [73,74]. Only isomiRs with at least 1 RPM in at least one of the experimental conditions were included for analysis.

**Dual-luciferase assay**

Primary rat cortical neurons were transfected in triplicates with 100 ng pmiRGlo luciferase reporters alone or with 10 nM miRNA mimics or LNA miRNA inhibitors. Neurons were lysed in 5× Passive Lysis buffer (Promega) 2 days after transfection, and dual-luciferase assay was performed using homemade reagents as previously described [75] on the GloMax R96 Microplate Luminometer (E1941, Promega).

**Quantification and statistical analysis**

Data are represented as mean ± standard deviation, unless otherwise stated. Three to four independent experiments were performed for each data set, as stated in the figure legends. For imaging experiments, approximately ten neurons were analyzed per experimental condition, in each experiment. $P$-values were calculated with two-tailed Student's $t$-test, homoscedastic (type 2) or heteroscedastic (type 3), as stated in the figure legends; $*P < 0.05$; $**P < 0.01$.

**Data availability**

Small RNA sequencing data: Gene expression Omnibus GSE106518.

**Expanded View** for this article is available online.

## Acknowledgements

We thank M Kiebler, C Hanus, W Fillipowicz, C Culmsee, M Buenemann, G Meister for kindly sharing reagents, and R Fiore for critical reading of the manuscript. A.A. was a recipient of postdoctoral research fellowships by the Alexander von Humboldt Foundation and EMBO non-stipendiary long-term fellowship. S.B. was a postdoctoral fellow of the von-Behring Rontgen Stiftung. Research in the laboratory of G.S. was funded by the DFG (SPP1738; FOR2107; SCHR 1136/8-1).

## Author contributions

Conceptualization, AA and GS; Data acquisition, AA, SB, and AI; Data analysis, AA and SK; Resources, RJ and GS; Writing and Editing, AA and GS; Funding acquisition, AA and GS.

## Conflict of interest

The authors declare that they have no conflict of interest.

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
