## [Review Process File · EMBO Reports]

The dynamic recruitment of TRBP to neuronal membranes mediates dendritogenesis during development

Anna Antoniou, Sharof Khudayberdiev, Agata Idziak, Silvia Bicker, Ralf Jacob and Gerhard Schratt

Review timeline:

Submission date:	18 July 2017
Editorial Decision:	21 August 2017
Revision received:	2 November 2017
Editorial Decision:	21 November 2017
Revision received:	22 November 2017
Accepted:	27 November 2017

Editor: Esther Schnapp & Martina Rembold

Transaction Report:

1st Editorial Decision

21 August 2017

Thank you for the submission of your research manuscript to our journal. We have now received the full set of referee reports that is copied below. Since Esther Schnapp is currently traveling I have temporarily taken over the handling of your manuscript.

As you will see, the referees acknowledge the potential interest of the findings but they also raise a number of issues and suggest experiments to strengthen the proposed role of miR-16 and its central importance to the phenotype upon TRBP/Dicer dissociation after BDNF treatment. Referee 1 also suggests to investigate the mechanism of TRBP redistribution in more detail and referee 2 proposes to use astrocytes instead of HEK293T cells as control to see if the mechanism is neuron-specific. Importantly, referee 3 is concerned that the observed changes upon BDNF-treatment are rather subtle and these concerns should be taken into account and discussed.

Given these constructive comments, we would like to invite you to revise your manuscript with the understanding that the referee concerns (as detailed above and in their reports) must be fully addressed and their suggestions taken on board. Please address all referee concerns in a complete point-by-point response. Acceptance of the manuscript will depend on a positive outcome of a second round of review. It is EMBO reports policy to allow a single round of revision only and acceptance or rejection of the manuscript will therefore depend on the completeness of your responses included in the next, final version of the manuscript.

Revised manuscripts should be submitted within three months of a request for revision; they will otherwise be treated as new submissions. Please contact us if a 3-months time frame is not sufficient for the revisions so that we can discuss the revisions further.

Supplementary/additional data: The Expanded View format, which will be displayed in the main HTML of the paper in a collapsible format, has replaced the Supplementary information. You can submit up to 5 images as Expanded View. Please follow the nomenclature Figure EV1, Figure EV2

etc. The figure legend for these should be included in the main manuscript document file in a section called Expanded View Figure Legends after the main Figure Legends section. Additional Supplementary material should be supplied as a single pdf labeled Appendix. The Appendix includes a table of content on the first page, all figures and their legends. Please follow the nomenclature Appendix Figure Sx throughout the text and also label the figures according to this nomenclature. For more details please refer to our guide to authors.

Regarding data quantification, can you please specify the number "n" for how many experiments were performed, the bars and error bars (e.g. SEM, SD) and the test used to calculate p-values in the respective figure legends? This information is currently incomplete and must be provided in the figure legends. Please also include scale bars in all microscopy images.

We now strongly encourage the publication of original source data with the aim of making primary data more accessible and transparent to the reader. The source data will be published in a separate source data file online along with the accepted manuscript and will be linked to the relevant figure. If you would like to use this opportunity, please submit the source data (for example scans of entire gels or blots, data points of graphs in an excel sheet, additional images, etc.) of your key experiments together with the revised manuscript. Please include size markers for scans of entire gels, label the scans with figure and panel number, and send one PDF file per figure or per figure panel.

As part of the EMBO publication's Transparent Editorial Process, EMBO reports publishes online a Review Process File to accompany accepted manuscripts. This File will be published in conjunction with your paper and will include the referee reports, your point-by-point response and all pertinent correspondence relating to the manuscript.

I look forward to seeing a revised version of your manuscript when it is ready. Please let me know if you have questions or comments regarding the revision.

REFeree REPORTS

Referee #1:

The work of Anna Antoniou et al., explores intracellular dynamics of Dicer cofactor, TRBP, during activity-dependent neuronal development. BDNF drives dissociation of TRBP from Dicer and a drop in miRNA-16 biogenesis which results in target expression and a change in dendritic spines morphogenesis.

major

(1) TRBP switch controlling its distribution between RER and cytoplasm is an important part of the molecular response mechanism. Characterize the molecular switch. Maybe it depends on Phosphorylation of TRBP?

(2) Why (BDNF) stimulation affects production of only microRNA-16. What allows such specificity? Might it be because of the AU rich sequence? (see *Cell*. 2005 120(5):623-34 PMID: 15766526). Could it be because of Wnt signaling crosstalk with BDNF through microRNA-16 (see *Nature* 449, 183-188 2007 | doi:10.1038/nature06100 and *Neuroreport*. 2012 Feb 15; 23(3): 189-194. doi: 10.1097/WNR.0b013e32834fab06).

(3) Regulation of ER activity might enable stimulus-dependent regulation of local membrane

composition at dendritic spines. Can the authors provide data or at least discuss the idea that TRBP or the regulated microRNA affect downstream local translation on / at the RER?

(4) Shorter isomicroRNAs (Fig 4) might have functional consequences. Is the seed modified? What targets gained / lost. Discuss the consequences.

minor

(5) Numbering of lines are missing

Referee #2:

SUBJECT: EMBOR-2017-44853-T

The study from Antoniou et al., reports a novel and neuronal specific mechanism for the regulation of pre-miRNA biogenesis via the dynamic localization of TRBP at ER membranes, which is necessary for BDNF-induced neuritogenesis. In particular, the authors show that this mechanism is responsible for the maturation of miR-16-5p and BDNF translation, a target of miR-16, and that this pathway in turn controls neurite complexity in growing cortical/hippocampal neurons.

Overall, the MS is well-written, the experiments are well designed and results generally support the conclusions. However, there are a few issues that in my opinion would need to be addressed. Please find specific comments/suggestions below.

In my opinion, if a pre- miRNA is processed locally, then both the precursor and the machinery required for processing should be present in the dendrites. To prove that pre-miR-16 is actually enriched in distal parts of the neuron, I would suggest performing in situ hybridization, or qRT-PCR on RNA isolated from dendrites.

To prove the neuronal-specificity of the local pre-miRNA maturation mechanism the authors use Hek293. The experiment fits the scope, however I believe it would be much more informative to check the presence of this mechanism in astrocytes, as proposed by the authors in the discussion.

It seems that the data shown in Fig 3b and 3 C are not fully internal consistent. Why after 20 minutes-BDNF is mature miR-16-5p down (deep seq, 3B), but not in qRT-PCR? (3C).

Concerning the investigation of functional the effect of this pathway in primary dendrites (figs 5 and 7): How are the dendrites identified? Perhaps the authors might consider changing "dendrites" to "neurites", or to use dendrite specific markers.

Referee #3:

In the present study, Antoniou et al. aim to characterize subcellular localisation of the RISC complexes in developing neurons and to investigate the potential function of ER associated RISCs in dendritic growth. Based on subcellular biochemical fractionation analysis and colocalization analysis via confocal microscopy, they demonstrated that several major RISC components including Ago1,2, Dicer1, PACT and TRBP are more enriched in membrane fractions, especially the ER membranes. This seems to be unique to neurons, as HEK 293 cells don't manifest the same characteristic. The authors further dissect the functional role of ER associated RISCs in dendritogenesis, and found that TRBP dissociates from Dicer and redistributes from ER to cytosol upon BDNF treatment, leading to a reduction of few specific miRNAs, namely mir16 family. As mir-16 also targets to BDNF, the BDNF dependent regulation of mir-16 forms an autoregulatory loop to potentiate dendritogenesis in developing neurons. Interestingly, small RNA seq results suggest that the reduction in mir-16 is caused by enhancement of miRNA trimming. Finally, to support the requirement of ER tethered TRBP for miRNA-16 biogenesis, , they used membrane bound TRBP to express in neurons. The interaction of Dicer and TRBP is increased, leading to a blockage of dendritogenesis.

The manuscript is nicely organized and this cell type specific regulatory mechanism of miRNA

biogenesis involving subcellular localization is interesting, although the novelty of ER associated RISC components is compromised due to similar observations were reported previously (Barman & Bhattacharyya, 2015; Li et al, 2013). Whilst the miRNA biogenesis components localised to the ER might be true, nearly all of the changes upon BDNF treatment are very subtle, I feel the current data are still not solid enough to support the conclusion.

Major points are listed below:

1. In fig 1D, the size of TRBP in cytosolic and membrane fractions is different. In addition, TRBP band looks different in 1D and 1E. Is it possible that different posttranslational modifications of TRBP might account for its subcellular localizations in neurons and HEK 293? Does this affect the conclusion of "neuron specific" ER localization? Quantification in 1D does not justify the WB result.
2. It has been reported that these (RISC) Proteins PACT, TRBP, and Dicer are present and active in nucleus (Gagnon et al, 2014), and can bind to specific pre-miRNAs. The authors should check if there is any nucleus contamination during sequential detergent extraction (Fig1A and S1A).
3. It is very difficult to distinguish the subtle colocalization changes in Fig2C and S2D, the authors may consider to add another panel depicting only those colocalization areas. Also, changes in Fig2D are very marginal, leading to the concern if this is physiologically relevant, given that BDNF can lead to many changes.
4. The authors suggest BDNF leads to weaken the interaction of TRBP/Dicer at ER thus changing the miRNA trimming process. Small RNA seq shows minimal shift of isoMir production. Then they went on using miR-16 sensor to test the loss-of-function of mir-16 during dendritogenesis. I think this does not actually justify this phenotype is due to the loss of mir-16 or the changes of isoMirs of mir-16 upon TRBP/Dicer dissociation after BDNF treatment.
5. In Fig 3A, the IP exp was conducted using total lysates, which raises the possibility that BDNF induced reduction in Dicer -TRBP interaction might also occur in cytosol. We suggest to perform an IP exp using cytosolic fraction to rule out this possibility.
6. The restoration of mir-16 by mGFP-TRBP expression should be included in Fig7. Also a sequential detergent extraction should be performed to confirm the ER localization of mGFP-TRBP.

Minor comments:

1. Fig1A figure legend is too short, making it difficult to catch what it means for each lane. Several quantifications are reflected by marginal change despite significant. In many cases, the standard deviation is huge (i.e. Fig1D, Fig1F.). More independent experiments need to be performed to minimize the technical variations.
2. Fig 6 seems to be a minor data to the conclusion of this paper, might not be appropriate to put in major figures.
3. Fig S7C, the sholl analysis of GFP expressing neurons look quite different from that of GFP-TRBP or mGFP-TRBP OE neurons, is there any explanations?

References

Barman B, Bhattacharyya SN (2015) mRNA Targeting to Endoplasmic Reticulum Precedes Ago Protein Interaction and MicroRNA (miRNA)-mediated Translation Repression in Mammalian Cells. *The Journal of biological chemistry* 290: 24650-24656

Gagnon KT, Li L, Chu Y, Janowski BA, Corey DR (2014) RNAi factors are present and active in human cell nuclei. *Cell reports* 6: 211-221

Li S, Liu L, Zhuang X, Yu Y, Liu X, Cui X, Ji L, Pan Z, Cao X, Mo B, Zhang F, Raikhel N, Jiang L, Chen X (2013) MicroRNAs inhibit the translation of target mRNAs on the endoplasmic reticulum in Arabidopsis. *Cell* 153: 562-574

1st Revision - authors' response

2 November 2017

Referee #1:

The work of Anna Antoniou et al., explores intracellular dynamics of Dicer cofactor, TRBP, during

activity-dependent neuronal development. BDNF drives dissociation of TRBP from Dicer and a drop in miRNA-16 biogenesis which results in target expression and a change in dendritic spines morphogenesis.

major

(1) TRBP switch controlling its distribution between RER and cytoplasm is an important part of the molecular response mechanism. Characterize the molecular switch. Maybe it depends on Phosphorylation of TRBP?

We have now further investigated the mechanisms underlying the redistribution of TRBP from the neuronal membrane to the cytosol upon BDNF treatment with several different experiments. Based on our results, we can rule out phosphorylation of known sites on TRBP as a central mechanism. First, the interaction of previously reported phosphorylation mutants of TRBP (S4A, S4D) with Dicer was indistinguishable from wild type TRBP in HEK293 cells (*Appendix Fig S3A*). Second, sequential detergent extraction experiments suggest that GFP-TRBP S4A and S4D have a similar cytoplasmic/membrane ratio as wild type TRBP (*Appendix Fig S3B*). Third, phosphatase treatment of neuronal extracts followed by Western blotting provided no clear evidence for the existence of phosphorylated TRBP variants in the neuronal cytoplasm (in contrast to the known phospho-protein Myosin Va, which was used as a positive control), as would have been expected if TRBP phosphorylation would trigger its release from the neuronal membrane (*Appendix Fig S3C*). While our data argue against an involvement of TRBP phosphorylation, other post-translational modifications (PTMs) could well be involved in the BDNF switch. However, determination of such PTMs would require challenging mass spec analysis, which is currently not set up in our laboratory and in our view beyond the scope of this manuscript.

In addition, we obtained preliminary evidence about the signalling pathways that are engaged downstream of BDNF. The intracellular calcium chelator BAPTA-AM completely blocked BDNF-dependent accumulation of TRBP in the cytosol of neurons (*Fig 2F*), suggesting that intracellular calcium is required for TRBP relocalization.

(2) Why (BDNF) stimulation affects production of only microRNA-16. What allows such specificity? Might it be because of the AU rich sequence? (see Cell. 2005 120(5):623-34 PMID: 15766526). Could it be because of Wnt signaling crosstalk with BDNF through microRNA-16 (see Nature 449, 183-188 2007 | doi:10.1038/nature06100 and Neuroreport. 2012 Feb 15; 23(3): 189-194. doi: 10.1097/WNR.0b013e32834fab06).

First of all, we would like to stress that not only production of miR-16 is affected by BDNF, but also the expression of several other canonical isomiRs, in particular those belonging to the miR-16 family (*Fig 4*). In addition, we observe quite profound changes in isomir distribution for a number of additional microRNAs (*Fig EV3*). Importantly, our observations are in line with a recent publication from the Kim lab (*see Cell Reports 2014, doi: 10.1016/j.celrep.2014.09.039*), where it was shown that TRBP knockout had very subtle effects on miRNA production, but affected the fidelity of pre-miRNA processing of a specific set of microRNAs. The mechanism underlying this selectivity however is still unknown, and its elucidation would require extensive biochemical characterization of pre-miRNA complexes in neurons under different stimulation conditions which is clearly beyond the scope of this report.

Following the reviewer's suggestion, we have inspected the sequences of miRNAs that display isomiR changes following BDNF stimulation (*Fig EV3*) and looked for the presence of AU-rich sequences. We found that only about a third of affected isomiRs (9 isomiRs out of 33 isomiRs shown in *Fig EV3*) have the UAAAU sequence, constituting around 40% of the canonical isomiRs that change with BDNF stimulation (*Fig 4A*). Further, we did not detect changes in the processing of pre-miR-195, which has the same mature miRNA sequence as miR-16-5p, whereas we detected changes in the processing of other miR-16 family members such as miR-15b-5p (*Fig 4*) and miR-15a-5p (*Fig EV3*), which do not bear AUUUA or an otherwise AU-rich sequence (around 60% AU content). We therefore speculate that this motif is unlikely to confer specificity in TRBP-mediated pre-miR-16 processing, although we cannot exclude the possibility that it is involved in later steps of translational repression (for example via TTP and Ago, *Cell 2005*). However, further investigation of this pathway is outside the scope of the current study.

We have now further discussed the relationship between BDNF/Wnt signalling and miR-16-5p. As pointed out by the reviewer, Martello *et al* (*Nature* 2007) showed that Wnt/ β -catenin signaling inhibits the expression of miR-15-5p and miR-16-5p via post-transcriptional mechanisms. Furthermore, BDNF expression is induced by Wnt signaling (*Neuroreport*, 2012), and previous publications from the Kellerman lab have shown that miR-16-5p antagonizes Wnt signaling by targeting the serotonin transporter and mediates adult neurogenesis following administration of SSRI anti-depressants (*Transl. Psy.* 2011, *Science* 2010). Interestingly, β -catenin, which mediates the transcriptional output of the canonical Wnt pathway, directly targets the Dicer1 gene and mediates stress resilience in mice (*Nature* 2014, doi:10.1038/nature13976), suggesting an intricate relationship between global miRNA maturation and canonical Wnt signaling. Notably, we did not detect changes in the levels of Dicer following the short BDNF stimulation protocol (*Fig EV1E*). We therefore argue that although further investigation of the role of Wnt signaling in TRBP-mediated miR-16-5p processing would be of interest, it is currently out of the scope of this study.

(3) Regulation of ER activity might enable stimulus-dependent regulation of local membrane composition at dendritic spines. Can the authors provide data or at least discuss the idea that TRBP or the regulated microRNA affect downstream local translation on / at the RER?

We now provide new data showing that pre-miR-16 is present in neuronal processes (*Fig 6D*), suggesting that TRBP-dependent processing of pre-miR-16 could at least in part occur locally in neuronal dendrites. In addition, we further discuss the implications of this mechanism for local mRNA translation at the rER and dendritic development.

(4) Shorter isomiRNAs (Fig 4) might have functional consequences. Is the seed modified? What targets gained / lost. Discuss the consequences.

Indeed, changes in Dicer cleavage position would lead to a shift in the seed region of 3p miRNA isomiRs. From our small RNA sequencing analysis we identified two 3p miRNAs whose non-canonical isomiRs are significantly affected by BDNF stimulation; a decrease in the miR-9a-3p +1 isomiR and an increase in miR-22-3p -1 isomiR. Following the suggestion of the reviewer, we performed target analysis of the shifted seeds (<http://www.targetscan.org>), followed by gene ontology (<https://david.ncicrf.gov>) of predicted targets in the rat genome. Firstly, we detected functional enrichment of miR-9a-3p +1 targets in the endocytic pathway (*Benjamini p-value 0.02*), unlike the canonical miR-9a targets. Secondly, the miR-22-3p -1 isomiR is predicted to target proteins with phosphatase activity (*Benjamini p-value 0.066*), whereas the canonical isomiR has endosomal trafficking-related targets (*Benjamini p-value 0.44*) (*Appendix Table S3*). Thus, the potential decrease in miRNA-mediated targeting of endocytic processes may reflect the requirement of lipid delivery to the plasma membrane during neuronal dendrite outgrowth. However, since template and non-template miRNA modifications affect the stability and turn-over of miRNAs, this model is highly speculative. Nevertheless, we have now included a discussion of potential shifted seed targets in the revised manuscript and we additionally highlighted the previously reported role of these miRNAs in nervous system development.

minor

(5) Numbering of lines are missing

Line numbers have been added.

Referee #2:

SUBJECT: EMBOR-2017-44853-T

The study from Antoniou et al., reports a novel and neuronal specific mechanism for the regulation of pre-miRNA biogenesis via the dynamic localization of TRBP at ER membranes, which is necessary for BDNF-induced neuritogenesis. In particular, the authors show that this mechanism is responsible for the maturation of miR-16-5p and BDNF translation, a target of miR-16, and that this pathway in turn controls neurite complexity in growing cortical/hippocampal neurons.

Overall, the MS is well-written, the experiments are well designed and results generally support the

conclusions. However, there are a few issues that in my opinion would need to be addressed. Please find specific comments/suggestions below.

In my opinion, if a pre-miRNA is processed locally, then both the precursor and the machinery required for processing should be present in the dendrites. To prove that pre-miR-16 is actually enriched in distal parts of the neuron, I would suggest performing in situ hybridization, or qRT-PCR on RNA isolated from dendrites.

As suggested by this referee, we have now included new data about the expression of pre-miR-16 from qPCR measurement of somatic and dendritic fractions of hippocampal neurons (Fig 6D), demonstrating that pre-miR-16 is present in neuronal processes, similarly to the dendritic pre-miR-341 and in contrast to the somatically-enriched pre-miR-137. This result is therefore consistent with a local component of this pathway.

To prove the neuronal-specificity of the local pre-miRNA maturation mechanism the authors use Hek293. The experiment fits the scope, however I believe it would be much more informative to check the presence of this mechanism in astrocytes, as proposed by the authors in the discussion.

We now show that TRBP is preferentially cytoplasmic in astrocytes (Fig 1E, Appendix Fig S1C), very similar to the results obtained from HEK293 cells (Fig 1F, Appendix Fig S1D). Therefore, we propose that membrane localization of TRBP is a specific feature of neurons, which could reflect the higher demand for the local regulation of protein synthesis at specific membrane compartments such as the rER.

It seems that the data shown in Fig 3b and 3 C are not fully internal consistent. Why after 20 minutes-BDNF is mature miR-16-5p down (deep seq, 3B), but not in qRT-PCR? (3C).

We agree that there is a discrepancy between the results obtained from qPCR and deep sequencing. Possible explanations are that qPCR only detects a single isomir (usually the most prevalent one listed in databases, such as MirBase), whereas in deep sequencing, expression of different isomirs is summed up in order to get an average expression value for a given microRNA. In addition, the sensitivity of our RNAseq is higher compared to qPCR, since we used a very large sequencing depth. Thereby, we might pick up more subtle changes in miRNA expression in RNAseq compared to qPCR. Consistently, the average levels of miR16 measured in qPCR were still decreased, albeit non-significantly, following 20 min of BDNF stimulation.

Concerning the investigation of functional the effect of this pathway in primary dendrites (figs 5 and 7): How are the dendrites identified? Perhaps the authors might consider changing "dendrites" to "neurites", or to use dendrite specific markers.

Given the differentiation stage of these neurons, we are actually convinced that the processes shown in Fig 5 and Fig 7 represent dendrites, even without performing an immunocytochemistry staining for dendritic markers (e.g. MAP2). These neurons usually contain a single very thin and largely unbranched axon, which can be easily distinguished from dendrites. However, if the reviewer strongly feels that we should change our wording to "neurites" instead, we would be prepared to make these changes in the revised manuscript.

Referee #3:

In the present study, Antoniou et al. aim to characterize subcellular localisation of the RISC complexes in developing neurons and to investigate the potential function of ER associated RISCs in dendritic growth. Based on subcellular biochemical fractionation analysis and colocalization analysis via confocal microscopy, they demonstrated that several major RISC components including Ago1,2, Dicer1, PACT and TRBP are more enriched in membrane fractions, especially the ER membranes. This seems to be unique to neurons, as HEK 293 cells don't manifest the same characteristic. The authors further dissect the functional role of ER associated RISCs in dendritogenesis, and found that TRBP dissociates from Dicer and redistributes from ER to cytosol upon BDNF treatment, leading to a reduction of few specific miRNAs, namely mir16 family. As mir-16 also targets to BDNF, the BDNF dependent regulation of mir-16 forms an autoregulatory loop to

potentiate dendritogenesis in developing neurons. Interestingly, small RNA seq results suggest that the reduction in mir-16 is caused by enhancement of miRNA trimming. Finally, to support the requirement of ER tethered TRBP for miRNA-16 biogenesis, , they used membrane bound TRBP to express in neurons. The interaction of Dicer and TRBP is increased, leading to a blockage of dendritogenesis.

The manuscript is nicely organized and this cell type specific regulatory mechanism of miRNA biogenesis involving subcellular localization is interesting, although the novelty of ER associated RISC components is compromised due to similar observations were reported previously (Barman & Bhattacharyya, 2015; Li et al, 2013). Whilst the miRNA biogenesis components localised to the ER might be true, nearly all of the changes upon BDNF treatment are very subtle, I feel the current data are still not solid enough to support the conclusion.

We appreciate that this referee finds our study nicely organized and interesting. We are aware of the mentioned papers, would like to stress however that they deal with Ago-dependent mRNA translation at the ER, not with localized pre-miRNA processing at the ER. To the best of our knowledge, our study is the first to describe growth-factor dependent regulation of microRNA biogenesis at the neuronal ER. With regard to the concern that the observed effects are very subtle and that the data set is not solid enough, we would like to refer to a number of publications that reported that BDNF effects with regard to neuromorphological parameters (e.g. dendrite growth) are usually in the range of 20% (*EMBO J. DOI 10.1038/emboj.2009.10; J. Neuroscience DOI: 10.1523/JNEUROSCI.0012-13.2013; EMBO Rep. DOI 10.15252/embr.201541218*). We have also calculated the number of independent experiments needed with regard to expected effect sizes and variation of the data. Using these criteria, we report statistically significant differences with an error probability of less than 5%. We therefore do not see the need for increasing the *n* from a statistical point of view. Finally, we have now added dashed lines to indicate fold-changes induced by BDNF more clearly.

Major points are listed below:

1. In fig 1D, the size of TRBP in cytosolic and membrane fractions is different. In addition, TRBP band looks different in 1D and 1E. Is it possible that different posttranslational modifications of TRBP might account for its subcellular localizations in neurons and HEK 293?

We have now checked a possible role of phosphorylation in TRBP relocalization, and did not obtain clear evidence for TRBP phosphorylation in neuronal cytoplasmic extracts (*Appendix Fig S3C*). We can also rule out an involvement of previously identified phosphorylation sites on TRBP (*Appendix Fig S3A-B, see also our comments to referee no.1 concerning this issue*). Nevertheless, this still leaves open the possibility that other PTMs (ubiquitylation, palmitoylation, etc.) are involved. Identification of such modifications however is clearly beyond the scope of this study.

Does this affect the conclusion of "neuron specific" ER localization?

Our statement of neuron-specific ER localization is based on our results from cellular fractionation experiments in neurons, HEK293T cells and now astrocytes (*Fig 1D-F, EV1A, Appendix Fig S1A&C-D*). These results clearly show that the vast majority of TRBP protein recognized by the employed antibody is membrane localized in neurons, but not HEK293T cells or astrocytes, irrespective of potential cell-type specific PTMs of TRBP. We have now included additional blots showing higher molecular weights of TRBP that may correspond to phosphorylation or other PTMs (*see also comments above*) are not present in the membrane fraction of astrocytes or HEK293T cells (*Appendix Fig S1C-D*) or in purified ER microsomes in neurons (*Appendix Fig S1A*).

Quantification in 1D does not justify the WB result.

In our opinion, the WB in 1D clearly shows the relative enrichment of TRBP and other complex components (Dicer, PACT) at neuronal membranes compared to the cytosol. This is reflected by values of 0.2-0.4 (*Fig 1D, right panel*) in the quantification of the fraction of a given protein in the cytosol compared to the membrane.

2. It has been reported that these (RISC) Proteins PACT, TRBP, and Dicer are present and active in

nucleus (Gagnon et al, 2014), and can bind to specific pre-miRNAs. The authors should check if there is any nucleus contamination during sequential detergent extraction (Fig1A and S1A).

We have now checked for nuclear contamination of our fractions by reprobing the WB with an antibody specific for the nuclear enzyme HDAC1 (Fig EV1A) and found that this protein was exclusively detectable in the nuclear fraction. This result argues against the possibility that the signal for PACT, TRBP and Dicer detected in the ER fraction is a result of nuclear contamination. Moreover, we did not detect endogenous TRBP or PACT in the nucleus of hippocampal neurons using immunocytochemistry (Fig 1B, 2B, Appendix Fig S2B).

3. It is very difficult to distinguish the subtle colocalization changes in Fig2C and S2D, the authors may consider to add another panel depicting only those colocalization areas.

We had in fact already included blow-ups of dendritic regions of the neurons in Fig 2C (now Fig 2B) of the original version of our manuscript to better illustrate changes in co-localization, in particular those happening in the dendritic compartment. We would like to stress however that changes in co-localization are in the order of 20% or less, consistent with our results from Western blot. Therefore, pictures showing a clearly visible difference in co-localization between the different conditions would likely not be representative and could lead to misleading conclusions.

Also, changes in Fig2D are very marginal, leading to the concern if this is physiologically relevant, given that BDNF can lead to many changes.

As already pointed out in one of our previous responses, quantitative changes often observed with BDNF are in the range of 20%, especially in co-localization analysis where there is intrinsically higher background signal than in western blotting. We could nevertheless confirm our results using two measures of co-localization (Pearson's and Mander's analysis), and two different ER markers (Fig 2B-D, EV1F-G, Appendix Fig S2). We are therefore confident that our immunocytochemistry results are specific. Furthermore, we have clearly demonstrated a physiological relevance of the BDNF-dependent TRBP re-distribution, using a TRBP mutant that exclusively localizes to the membrane compartment (Fig 7, EV5).

4. The authors suggest BDNF leads to weaken the interaction of TRBP/Dicer at ER thus changing the miRNA trimming process. Small RNA seq shows minimal shift of isoMir production. Then they went on using miR-16 sensor to test the loss-of-function of mir-16 during dendritogenesis. I think this does not actually justify this phenotype is due to the loss of mir-16 or the changes of isoMirs of mir-16 upon TRBP/Dicer dissociation after BDNF treatment.

While we do see changes in the isomir distribution upon BDNF treatment only in a subset of miRNAs (Fig 4 and Fig EV3), these changes are highly significant and very consistent with the reported function of TRBP in assuring processing fidelity of the Dicer pre-miRNA complex (Kim et al., 2014, Cell Reports, doi: 10.1016/j.celrep.2014.09.039). In order to test whether reduced production of functional miR-16 isomirs in the BDNF condition is actually required for increased dendritic complexity, we re-introduced miR-16 into neurons using transfection of miR-16 mimics and found, as expected, that it completely blocked BDNF-induced dendritogenesis (Fig 5F-G, EV4C-D). In addition, we performed dendritogenesis assays in neurons expressing a Dicer binding-deficient TRBP truncation mutant (GFP-TRBP Δ) and show that similarly to miR-16 inhibition (Fig 5D-E, EV4A-B), and in contrast to mGFP-TRBP (Fig 7C-D, EV5C) or miR16 overexpression, GFP-TRBP Δ expression increases dendritic complexity and occludes BDNF-induced dendritogenesis (Fig 5H-I, EV4E-F). Therefore, our results are highly consistent with the proposed model, whereby TRBP dissociation from membrane-associated Dicer induces dendrite complexity via inhibition of miR-16 production.

Nevertheless, we agree that we do not directly address an involvement of miR-16 isomir shift, as opposed to reduced miR-16 levels, in the dendritogenesis phenotype. In fact, these two mechanisms could even be coupled, since it is well documented that trimming of miRNAs can affect the stability and thereby indirectly the expression levels of that miRNA. The assessment of individual isomirs however is technically challenging since it would require a miR-16 knockout background in which one could assess potentially different efficacies of miR-16 isomirs. We do not have such a model at

our predisposition and therefore think that such experiments are beyond the scope of the present study.

5. In Fig 3A, the IP exp was conducted using total lysates, which raises the possibility that BDNF induced reduction in Dicer-TRBP interaction might also occur in cytosol. We suggest to perform an IP exp using cytosolic fraction to rule out this possibility.

Following the suggestion of this reviewer, we have now included co-IP experiments with cytosolic extracts obtained from mock and BDNF-treated neurons (Fig EV2B). We did not observe any changes in the cytosolic Dicer-TRBP interaction, arguing that this effect is specific to neuronal (ER) membranes.

6. The restoration of mir-16 by mGFP-TRBP expression should be included in Fig7.

We included a miR-16 sensor assay (Fig 7B), which shows that expression of mGFP-TRBP fully restores miR-16 repressive activity in the context of BDNF. qPCR is challenging since it would require full transfection which can only be achieved by viral constructs. The latter however is problematic for TRBP, given that such constructs regularly exceed the viral packaging size due to the size of the TRBP protein.

Also a sequential detergent extraction should be performed to confirm the ER localization of mGFP-TRBP.

We have already provided results from Western blot (Fig EV5A) and immunocytochemistry (Fig EV5B) experiments with mGFP-TRBP transfected cells in the supplementary part of our manuscript that clearly show a preferential membrane localization of mGFP-TRBP.

Minor comments:

1. Fig1A figure legend is too short, making it difficult to catch what it means for each lane. Several quantifications are reflected by marginal change despite significant. In many cases, the standard deviation is huge (i.e. Fig1D, Fig1F.). More independent experiments need to be performed to minimize the technical variations.

As pointed out already in our previous comments, results are significant after considering appropriate number of replicates due to the estimated variations and effect sizes. Therefore, we do not see the need to replicate the experiments in order to reduce biological variability.

2. Fig 6 seems to be a minor data to the conclusion of this paper, might not be appropriate to put in major figures.

We would like to keep this data in the main part of the manuscript since it provides evidence that BDNF is a direct target of miR-16, which is an important conclusion of this study.

3. Fig S7C, the sholl analysis of GFP expressing neurons look quite different from that of GFP-TRBP or mGFP-TRBP OE neurons, is there any explanations?

As shown in Fig 7, expression of mGFP-TRBP completely blocks BDNF-dependent dendritic growth as assessed by the calculation of the total number of intersections obtained with Sholl analysis. It is therefore not surprising that the more detailed Sholl profiles of the different TRBP transfections also look quite different (Fig EV5C).

References

Barman B, Bhattacharyya SN (2015) mRNA Targeting to Endoplasmic Reticulum Precedes Ago Protein Interaction and MicroRNA (miRNA)-mediated Translation Repression in Mammalian Cells. *The Journal of biological chemistry* 290: 24650-24656

Gagnon KT, Li L, Chu Y, Janowski BA, Corey DR (2014) RNAi factors are present and active in human cell nuclei. *Cell reports* 6: 211-221

Li S, Liu L, Zhuang X, Yu Y, Liu X, Cui X, Ji L, Pan Z, Cao X, Mo B, Zhang F, Raikhel N, Jiang L, Chen X (2013) MicroRNAs inhibit the translation of target mRNAs on the endoplasmic reticulum in Arabidopsis. Cell 153: 562-574

2nd Editorial Decision

21 November 2017

Thank you for the submission of your revised manuscript to our journal. We have now received the enclosed reports from the referees and I am happy to tell you that all support its publication now. Only referee 3 still has some minor suggestions that I would like you to address before we can proceed with the official acceptance of your manuscript.

I look forward to seeing a final version of your manuscript as soon as possible. Please let me know if you have any questions or comments.

REFEREE REPORTS

Referee #1:

The new version is satisfying and I have no additional comments.

Referee #2:

SUBJECT: EMBOR-2017-44853V2

Comments for the authors

Overall the authors have addressed most of my concerns. I have no further remarks.

Referee #3:

The revised manuscript addressed concerns raised by the reviewers, including 1) examination of TRBP subcellular localization in astrocyte to support specific function of membrane-associated TRBP in neurons; 2) investigating the role of phosphorylation in TRBP localization; 3) providing potential calcium-dependent machinery in regulating TRBP subcellular localization; 4) ruling out the possible nuclear contamination in sequential extraction by including the nuclear marker HDAC; 5) determining the presence of pre-mir16 in dendrites; 6) further demonstrating that neurons expressing TRBP mutant deficient of Dicer binding can increase neurite outgrowth; 7) analysis of AU rich sequences in BDNF affected isomiRs; 8) discussion about the implications of this mechanism for local mRNA translation during dendritic development and the potential effect of a shift in the seed region of isomiRs (miRNA from 3p).

I agree with the authors' point that BDNF induced neuromorphological changes are in the range of 20%, and the combination of their immunostaining, biochemical fractionation and effects of different TRBP forms do support the conclusion. Overall I think the improved manuscript is interesting and brings certain degree of new insights to the field.

However, I have concerns about the summary that phosphorylation of TRBP is not involved in its subcellular localization control. In particular, from the lambda phosphatase treatment exp (Fig S3C), I can not see the phosphatase working to promote accumulation of unphosphorylated myosin V. If the assay did not work well in the positive control, it can not support the conclusion that TRBP is not phosphorylated in neurons. I also suggest the authors put arrows to clearly point out the major band of TRBP (supposed to be non-modified TRBP) in the WB, as well as the phosphorylated- and unphosphorylated myosin Va.

2nd Revision - authors' response

22 November 2017

Concerning the remaining point raised by referee 3, we have marked the bands of phosphorylated and non-phosphorylated myosin V in the blot (Fig. S3C) with arrows. Since the lower band of myosin V has different migratory behaviour dependent on the presence or absence of a phosphatase inhibitor, this serves as a positive control for our assay. We therefore do not see the necessity to change our conclusions regarding TRBP phosphorylation in neurons.

YOU MUST COMPLETE ALL CELLS WITH A PINK BACKGROUND ↓
PLEASE NOTE THAT THIS CHECKLIST WILL BE PUBLISHED ALONGSIDE YOUR PAPER

Corresponding Author Name:
Journal Submitted to:
Manuscript Number: